# Pharmaceutical Supply Chain in China: Pricing and Production Decisions with Price-Sensitive and Uncertain Demand

**Suhan Wu [1] [ID], Min Luo [2],* [ID], Jingxia Zhang [3], Daoheng Zhang [4] and Lianmin Zhang [5]**

[1] School of Economics and Management, Nanjing Polytechnic Institute, No. 188 Xinle Road, Neo Jiangbei District, Nanjing 210048, China; wusuhan@njpi.edu.cn

[2] School of Management, Shenzhen Institute of Information Technology, No. 2188 Longxiang Avenue, Longgang District, Shenzhen 518172, China

[3] School of Information Science and Engineering, Jinling College Nanjing University, No. 8 Xuefu Road, Pukou District, Nanjing 210089, China; zhangjingxia_127@163.com

[4] School of Engineering and Information Technology, University of New South Wales, Canberra, ACT 2600, Australia; alan.zhang1@adfa.edu.au

[5] Shenzhen Research Institute of Big Data, Daoyuan Building, No. 2001 Longxiang Avenue, Longgang District, Shenzhen 518172, China; lmzhang@sribd.cn

\* Correspondence: luom@sziit.edu.cn

**Abstract:** In this paper, we apply game theory to study the price competition between drugstores and hospitals in China's pharmaceutical supply chain. Motivated by drug shortages and price disparity problems, we build a simplified model with one supplier, one hospital, and one drugstore in which the sellers sell one kind of drug and compete on price. The hospital receives a discount from the government when ordering the drug and both sellers face a price-sensitive and uncertain demand. The existence and uniqueness of a Nash equilibrium are proved and closed-form solutions are found for linear demand cases. We characterize the pricing and ordering decisions of the hospital and drugstore. The analysis shows that high ex-factory price, high price sensitivity, and a small discount are three factors contributing to drug shortages. We consider two special kinds of linear demand to obtain insights into the drug price disparity problem.

**Keywords:** pharmaceutical supply chain; pricing; game theory; price-sensitive and uncertain demand





## 1. Introduction

In 1987, the World Commission on Environment and Development proposed the concept of "sustainable development", which refers to development that meets the needs of the present without compromising the ability of future generations to meet their own needs [1]. This statement declares that sustainable development is a balanced strategy among profitability, environmental protection and social responsibilities [2]. It involves many aspects of human activities which have brought ever-lasting awareness among government, industry, and the general public [3]. As a system involved in supplying products or services to consumers [4], the sustainable performance of a supply chain has been discussed in recent years. Facing the challenge of environmental dynamism, various collaboration mechanisms are needed in industrial supply chains to achieve sustainable goals such as emission peak and carbon neutrality, which are urgently required in developing economies [5–8].

Compared to other industries, the pharmaceutical industry is not only responsible for the development and manufacturing of medications, but also improving healthcare access. Thus, apart from pursuing maximum profits, the accessibility of medical products is also taken into consideration for participants in the pharmaceutical supply chain. This study mainly focuses on China's pharmaceutical supply chain. In 2000, China had 16,318 public hospitals [9]. This number increased rapidly to reach 37,000 in 2020. The growing need for healthcare also led to a thriving pharmaceutical industry. At the end of 2020, the number of medical manufacturer has reached 8170, with an annual gross output of 228.682 billion yuan [10]. In response to the development of the pharmaceutical industry,

China's government has promoted the reformation of the pharmaceutical supply chain. Traditionally, pharmaceutical retailers at different levels purchased drugs from different wholesalers, so the supply chain lacked competitive mechanisms, resulting in bureaucratic behavior, inefficiencies, and an imbalanced supply [11]. In the reformed pharmaceutical supply chain (see Figure 1), drug manufacturing firms can directly sell medical products to hospitals and drugstores [12]. This reformation has gradually changed the supply chain from a government-controlled system to a market-oriented one, which has improved its flexibility and reliability.

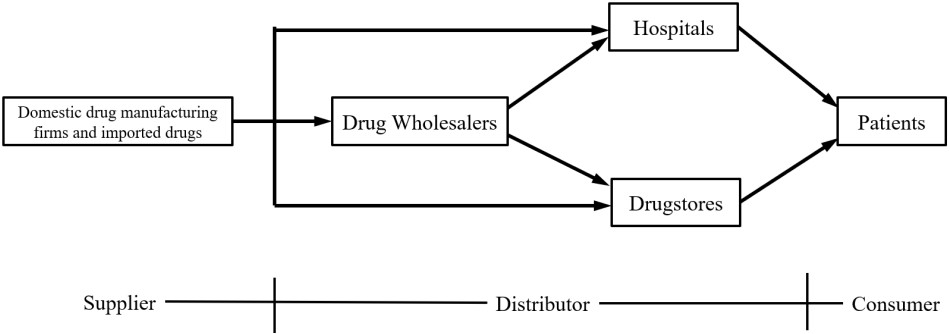

**Figure 1.** Reformed pharmaceutical supply chains in China.

### 1.1. Motivations

Despite the improvements in China's pharmaceutical chain, some problems have occurred in recent years. A social media user called *Renmin Zixun* reported that some drugs were out of stock in hospitals [13]. The situation was even worse when methimazole, which is the first choice for treating hyperthyroidism, began to go out of stock in some major cities [14]. In addition to shortages, the price gap between drugstores and hospitals for the same kinds of drugs is also noteworthy. Due to the lower operational cost, drugstores often sell drugs at lower prices than hospitals do. However, in some cases a drugstore's selling price can be four or five times as expensive as that of a hospital [15]. This can make drugs unaffordable for people who do not have time to see a doctor, which has led to several complaints from the public.

Motivated by such issues, we address the following questions in this paper:

- How can a mathematical model be developed to depict pricing and production decisions in China's pharmaceutical supply chain with price-sensitive and uncertain demand?
- How do factors such as price sensitivity, ex-factory cost or governmental discounts influence the optimal decisions of drug retailers?
- Which factors have contributed to the drug shortage problem?
- Why do drugstores sometimes charge much more for a drug than hospitals do?

Ref. [16] has provide an excellent discussion of the newsvendor pricing game, we apply their methodologies to depict the competition between drugstores and hospitals in China's pharmaceutical supply chain. In this paper, we consider the price competition between one hospital and one drugstore that obtain a single kind of drug from one drug wholesaler. Each of the two retailers faces a price-sensitive and uncertain demand and needs to decide both the order quantity and price to maximize its profit. As public hospitals in China are often state-owned and regarded as social welfare agencies, they may receive government discounts when purchasing drugs.

This study derives a strategic model to analyze the drug shortage and price disparity problems. The research objectives could be stated as follows:

- Describing the drug pricing competition between drugstores and hospitals in China's pharmaceutical supply chain.

- Obtaining the optimal strategies of participants in pharmaceutical supply chain and analyzing the influential factors of optimal prices, order quantities, profits and satisfaction rates.
- Identifying the main reasons for drug shortage and price disparity problems in China and providing suggestions for solving the problems.

### 1.2. Contributions

In this study, we investigate the existence and uniqueness of the pricing equilibrium and obtain the closed form of optimal prices in a linear demand case. This result depicts the pricing strategies of hospitals and drugstores and thus provides some insights into drug shortage and price disparity problems.

The main contributions of this paper are as follows:

- Building a pricing model in China's pharmaceutical supply chain with price-sensitive and uncertain demand considering governmental discount.
- Proving the existence and uniqueness of a pure-strategy Nash equilibrium in the game and deriving the closed form of two sellers' optimal prices under the assumption of linear uniformly distributed demand.
- Analyzing the impacts of ex-factory price and government discounts on optimal prices and satisfaction rates to obtain insights on the drug shortage problem.
- Analyzing the impacts of price sensitivity on optimal prices, order quantities, expected profits, and satisfaction rates in two special cases of linear demand to provide insights into the price disparity problem.
- Providing suggestions for how the government can act to avoid drug shortage and price disparity problems.

After elaborating on the motivations and contributions of our research, the remainder of this paper is organized as follows. In Section 2, we review some important works that describe the present situation of China's pharmaceutical industry and the strategic newsvendor model that we use in our analysis. Section 3 presents research methods of the paper together with model descriptions and assumptions. Section 4 contains model formulation and equilibrium analysis. Two kinds of linear demand function are discussed and some insights into the drug shortage and price disparity problems are provided. Section 5 provides a numerical analysis of the results in Section 4 and further explains the causes of the problems. Section 6 presents our conclusions, implications and some suggestions for future research.

## 2. Literature Review

A typical pharmaceutical supply chain may contain some or all of the following parts: primary manufacturers, secondary manufacturers, market warehouses (distribution centers), wholesalers, and retailers [17]. Only a few studies have considered the pharmaceutical supply chain in a particular country or region. Ref. [12] discussed the performance and distortions of the pharmaceutical market in China's health system reform. They concluded that the key factor in market and government failures is that all suppliers prefer higher-than-cost drugs, a problem that could be solved by introducing a new drug pricing mechanism. Ref. [18] analyzed the impact of radio frequency identification in Asia and Europe, leading to a better understanding of customer needs and buyer behavior in the context of pharmaceutical suppliers. Ref. [19] studied the sustainable performance of Ethiopian healthcare supply chain. Through a modeling approach, the bottlenecks of environmental supply chain could be identified. Other studies have mainly focused on the operational aspects of general pharmaceutical supply chains, which can be divided into long-term, mid-term, and short-term decisions [20]. Based on the research objectives, this sections mainly reviews the operations research of pharmaceutical supply chain and different approaches on newsvendor problem.

*2.1. Long-Term Decision Problems on Pharmaceutical Supply Chain*

Long-term decision problems, also known as strategic issues, mainly concern supply chain network design and capacity planning [21]. Pharmaceutical supply chain network design considers the co-ordination of participants in different countries. Mixed integer linear programing (MILP) models have been established to investigate problems such as plant operations [22], capacity-expansion planning [23], and product allocation or distribution [24,25]. Various assumptions were made in these studies, such as continuous time [22], the uncertainty of data and outsourcing of production [23], and different distribution costs and tax rates in different locations [25]. Motivated by these works, other researchers have established bi-objective MILP (BOMILP) models to handle pharmaceutical supply network design problems with uncertain parameters [26,27]. As the problems are often too large to be tractable in a reasonable time, researchers have often used decomposition algorithms to obtain acceptable results. For the capacity planning problem, ref. [28] described a deterministic model for allocating newcomers to existing sites. They claimed that taxation can have a huge effect on location decisions. Ref. [29] extended their work with a systematic programming approach to handle the situation of long-term, multi-site capacity planning under uncertainty and established a hierarchical algorithm to deal with the large-scale MILP problem. Ref. [30] built a stochastic model to describe the problem of clinical trials. They assumed that drugs went through different stages in their life cycles and trials of products would be completed at different times. They solved a four-product problem but their approach was limited by the complexity of the model. Refs. [31,32] considered the capacity planning problem of active pharmaceutical ingredients and final drug products. With uncertain clinical trial outcomes, they proposed an MILP formulation to ensure production capacity to meet uncertain demands and a decomposition algorithm for industrial-scale problems.

*2.2. Short-Term Decision Problems on Pharmaceutical Supply Chain*

Studies of short-term decisions have mainly included scheduling or sequencing decision problems, in which the decision-maker needs to decide a combination of variables, such as assignment to production lines, the number and size of batches, the length of campaigns, and material flows [20,33]. Ref. [34] considered the production planning and scheduling problem in multi-purpose batch chemical plants that can accommodate different products in many ways. Given the production requirements, they established a computer program to obtain production strategies and allocation times. Ref. [35] considered the scheduling problem of multistage batch plants, in which the optimal production policy to satisfy the demands for different products must be found before due dates. They proposed an MILP model with continuous time and established two solution strategies to solve it in a reasonable time. As realistic sizing problems are often too large to be tractable, researchers have turned their attention to the techniques for solving such problems. Ref. [36] summarized techniques that are useful for the design, planning, and scheduling of batch processes. They analyzed the performance of different approaches to solving MILP and MINLP problems in various contexts and gave some examples to illustrate them. For a large-scale MILP model for scheduling chemical batch processes, ref. [37] proposed an LP-based heuristic algorithm to reduce the size of the problem for which the optimal solution can be obtained within reasonable CPU-time.

*2.3. Mid-Term Decision Problems on Pharmaceutical Supply Chain*

The main considerations of mid-term decisions in pharmaceutical supply chains are product portfolio selection and inventory control problems [20]. For the first problem, if a pharmaceutical company wants to achieve sustainable development, it is necessary for it to manage its R&D processes. Thus, the choice of which products should be included in development projects is of great importance. Ref. [38] built a stochastic optimization model to depict pharmaceutical R&D processes, in which the development of new drugs is controlled by a series of continuation/abandonment options that determine whether

to proceed with development. They proposed a framework to guide the decisions in real cases and showed that the value of the abandonment option increased as market uncertainty increased. With the objective of maximizing profits at an acceptable level of risk, ref. [39] proposed a portfolio management approach to decide which projects should be selected. They established a probabilistic network model to depict all of the activities and resources involved in developing a new drug and built a genetic algorithm-based search for the optimal sequence with product dependencies and limited resources. Instead of finding exactly one group of decisions under certain conditions, ref. [40] concentrated on improving the quality of pharmaceutical resource management decisions and practices in the pharmaceutical R&D pipeline. They proposed a simulation–optimization framework to simulate the pharmaceutical workflow process. This framework contained both a resource manager and a strategy learner, and it obtained and improved scheduling and resource allocation control policies by learning from optimization agents. Ref. [41] considered another important aspect of the pharmaceutical R&D pipeline: the clinical trial planning problem. Given a portfolio of potential drugs and limited resources, they proposed a multi-stage stochastic programming formulation to decide which trails to perform in each period. They used a reduced set of scenarios to reduce the size of the problem and proposed a branch cut algorithm in a subsequent study [42]. The main ideas and approaches to product portfolio selection are well summarized in [43].

Another aspect of mid-term decisions in pharmaceutical supply chains is the inventory control problem. Different from the common situation, inventory control in a pharmaceutical supply chain may require a high level of customer service to manage perishable products [44]. To depict the situation for an inpatient hospital, ref. [45] considered a two-stage inventory management problem with perishable raw materials and finished goods in each stage. They developed a Markov decision process to decide optimal inventory and production policies for both stages and applied the framework to the drug Meropenem. Ref. [46] established a model with one company and one hospital, in which production and distribution were continuously reviewed and multiple products, variable lead times, and permissible payment delays were considered. To minimize the total cost, they developed a procedure to determine optimal solutions with constraints on space availability and customer service level, which could be used as a decision support tool. Instead of finding solutions for a general model, ref. [47] studied a hospital's inventory policy in detail. They established a multi-product $(s, S)$ model to obtain optimal allocations. They also claimed that the expected number of daily refills, the service level, and storage space utilization could be key performance indicators of tactical decisions that could be used to analyze the tradeoffs among the refill workload, emergency workload, and variety of drugs offered.

### 2.4. Newvendor Problem with Different Approaches

In this paper, a model based on the newsvendor problem is established to depict price competition between hospitals and drugstores. When reviewing the literature on pharmaceutical supply chain analysis, we did not find any studies similar to ours, but some studies have considered the newsvendor problem. We review some of them in the rest of this section.

The origin of the newsvendor problem can be traced to the 19th century [48]. It originates from a decision problem relating to how many newspapers a newsboy would order when facing a market with random demand. If he ordered too many, some newspapers would be discarded, but if the order quantity was too small, some customers would not get their newspapers, causing a loss of profit. Given the distribution of demand and selling price, the problem can be solved. Its closed-form solutions and conclusions were summarized by [49].

As the newsvendor model is widely used in operations management and applied economics, it is natural for researchers to consider it with different assumptions and constraints. Several researchers have discussed models with different demand functions. Ref. [50] assumed that the deterministic part of demand was a decreasing function of price and obtained closed-form solutions for a special kind of symmetrically distributed demand.

Ref. [51] considered a similar demand function with constant variance and explained why uncertainty often leads to a lower optimal selling price. Ref. [52] considered two more detailed structures of price dependent demands—the linear demand and the empirical demand—and proposed algorithms to find solutions.

Some other studies have considered sellers with different profit functions. The most representative of these is [53]. The authors changed the objective of sellers from "maximizing profit" to "maximizing the probability of achieving a certain profit" and worked out the closed-form solutions of the optimal order quantity with exponentially distributed demand.

Different constraints have also been considered by researchers. Ref. [54] considered the multi-product problem with a constraint that requires the seller to have stock not lower than a certain level. In follow-up work, ref. [55] provided four efficient algorithms to solve the single-period problem with one constraint.

Researchers have also studied price competition in the newsvendor model. Ref. [56] studied the interaction between the newsvendor model and game theory. They modeled the problem with competition among $N$ retailers, each facing a random demand, and analyzed the performance of a competitive decentralized supply chain, concluding that with an appropriate contract, a decentralized supply chain can act as well as a centralized chain. Ref. [16] proved the existence and uniqueness of equilibrium in a competitive game and showed that competition leads to a lower selling price at the equilibrium point by comparing it with a cooperative relationship between suppliers and sellers.

In addition to the classic research on the newsvendor problem, researchers have combined traditional methods with new techniques. Ref. [57] considered a data-driven newsvendor problem in which the demand was drawn from a random, independent sample. They analyzed the sample average approximation (SAA) approach and claimed that the demand distribution's weighted mean spread affects the accuracy of the SAA heuristic. To deal with a newsvendor problem in which the probability distribution of demand is unknown, ref. [58] proposed a deep learning algorithm in which demand forecasting and inventory-optimization were integrated. In numerical experiments, their algorithm was able to run without knowledge of the demand distribution and outperformed other approaches.

### 2.5. Research Gap

For operations research in the pharmaceutical supply chain, LP-based models such as MILP [22–25,29,31,32,35] or BOMILP [26,27] are established to investigate long-term and short-term decision problems such as network design, capacity planning and scheduling. For mid-term decision problems, stochastic optimization models [38,39,41] or decision processes [45,46] are applied to investigate product portfolio selection and inventory control problems, respectively.

This paper aims to investigate the pricing and production decisions of drugstores and hospitals in China's pharmaceutical supply chain, which belongs to the inventory control problem. In this area, researchers are mainly focused on optimal strategies of the hospital with perishable pharmaceutical products [45,47], or the interaction between upstream and downstream participants in the pharmaceutical supply chain [46]. However, few studies have considered the pricing competition between pharmaceutical retailers.

The innovation of this paper is mainly in applying game theory to a newsvendor model in China's pharmaceutical supply chain, focusing on the perspective of pricing and production decisions with price-sensitive and uncertain demand. In addition to the optimal prices and quantities, this paper also focuses on the drug shortage and price disparity problems in China and studies their contributing factors, corresponding with some managerial implications.

### 3. Research Methods

This paper applies game theory to study the price competition between retailers in China's pharmaceutical supply chain. The methodology is interpretivism as a newvendor pricing model with uncertain demand is established to depict the interaction between

hospitals and drugstores. The methods of this study involves four produces. Firstly, it describes the problem and presents the assumptions. Secondly, it derives the model formulation to get the optimal order quantities of participants in pharmaceutical supply chain and analyzes the equilibrium by game theory. Thirdly, two special kinds of linear demands are discussed and the closed-form of pricing decisions are derived to depict the competition between pharmaceutical retailers. Lastly, it numerically analyzes the influence of ex-factory prices, governmental discounts and market sensitivity on optimal prices, profits, order quantities and satisfaction rates. The results provide some insights in drug shortage and price disparity problems and some managerial implications in pharmaceutical supply chain.

*3.1. Problem Description*

From Figure 1 we see that there are two types of drug distribution in China. One is from domestic drug manufacturing firms to wholesalers, who then distribute the drugs to hospitals and drugstores. The other is directly from manufacturing firms to hospitals and drugstores. In both cases, patients obtain their drugs from hospitals or drugstores. Without loss of generality, we could merge the manufacturing firms and wholesalers in Figure 1 into one to simplify the problem (see Figure 2).

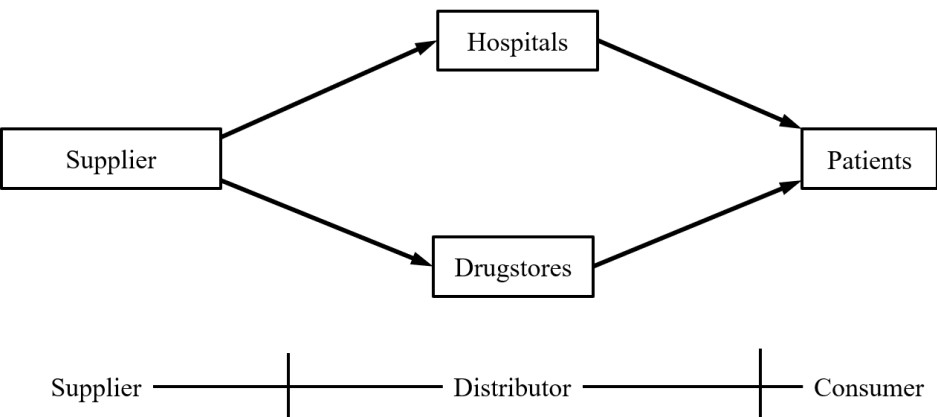

**Figure 2.** Simplified pharmaceutical supply chains in China.

To concentrate on the key issues of the problem, we consider a simplified supply chain of one hospital and one drugstore, where the objective of each player is to maximize its profit. Each of them sells only one kind of drug with price-sensitive and uncertain demand. Drugs are provided by a single supplier with infinite production capacity. The decision variables of the sellers include order quantity and selling price. However, the supplier decides the ex-factory price, which is exogenous for the distributors.

We further assume that patients have only two choices: one is to see a doctor at the hospital and obtain drugs there, the other is to go to the drugstore directly to buy the drug. The hospital, acting as a social welfare agency, receives a price discount from the government.

In our model, the sellers have the same goal to maximize their total profit. The process is as follows: first, the pharmaceutical supplier sets the ex-factory price of the drug; then, facing random demand, the two sellers simultaneously first decide their order quantities and then their selling prices [59].

Our notations are listed in Table 1:

**Table 1.** Model notations.

| Symbol | Description |
|---|---|
| $c$ | Ex-factory price for the supplier |
| $\phi$ | Discount factor for the hospital, where the hospital gets a marginal cost of $\phi c$, $\phi \in (0, 1]$ |
| $p_d$ | Selling price for the drugstore |
| $p_h$ | Selling price for the hospital |
| $Q_d$ | Order quantity for the drugstore |
| $Q_h$ | Order quantity for the hospital |
| $p_d^*$ | Optimal selling price for the drugstore |
| $p_h^*$ | Optimal selling price for the hospital |
| $Q_d^*$ | Optimal order quantity for the drugstore |
| $Q_h^*$ | Optimal order quantity for the hospital |
| $A_d$ | Reliability factor of the drugstore |
| $A_h$ | Reliability factor of the hospital |
| $D_d$ | Deterministic part of the drugstore's demand, also represented as $D_d(p_d, p_h, A_d)$ |
| $D_h$ | Deterministic part of the hospital's demand, also represented as $D_h(p_d, p_h, A_h)$ |
| $e_d$ | Price elasticity of the drugstore's demand |
| $e_h$ | Price elasticity of the hospital's demand |
| $\xi_d$ | Random part of the drugstore's demand |
| $f_{\xi_d}$ | Density function of $\xi_d$ |
| $F_{\xi_d}$ | Cumulative distribution function of $\xi_d$ |
| $\xi_h$ | Random part of the hospital's demand |
| $f_{\xi_h}$ | Density function of $\xi_h$ |
| $F_{\xi_h}$ | Cumulative distribution function of $\xi_h$ |
| $r_d^*(p_h)$ | Best response function of the drugstore |
| $r_h^*(p_d)$ | Best response function of the hospital |
| $RD_d$ | Drugstore's demand, where $RD_d = D_d \cdot \xi_d$ |
| $RD_h$ | Hospital's demand, where $RD_h = D_h \cdot \xi_h$ |
| $\Pi_d$ | Expected payoff of the drugstore, also represented as $\Pi_d(p_d, p_h, Q_d)$ |
| $\Pi_h$ | Expected payoff of the hospital, also represented as $\Pi_h(p_d, p_h, Q_h)$ |
| $\Pi_s$ | Expected payoff of the supplier, also represented as $\Pi_s(c)$ |

This model is a static non-cooperative game with pricing, where each of the players can make his decision only once. The two common ways to depict random demands with pricing are the additive form defined by $RD(p, \xi) = D(p) + \xi$ [51] and the multiplicative form defined by $RD(p, \xi) = D(p) \cdot \xi$ [60]. The demand is described as a deterministic part (decreasing with $p$) adds (or multiplies) a random part $\xi$. In our model, we use the multiplicative form to describe the demand.

*3.2. Assumptions*

In this section, we state the assumptions of our basic model and provide comments on some of them. Assumptions 1 and 3–5 are demand related assumptions which also appear in [16,56].

**Assumption 1.** *Assumptions on the random variable $\xi_d$ and $\xi_h$:*

1. *$\xi_d$ and $\xi_h$ are independent of $p_d$ and $p_h$, furthermore, $E[\xi_d] = E[\xi_d] = 1$. So $E[RD_d] = D_d(p_d, p_h, A_d)$, $E[RD_h] = D_h(p_d, p_h, A_h)$.*
2. *$\xi_d$ and $\xi_h$ are uniformly distributed on $[1 - \sigma_d, 1 + \sigma_d]$ and $[1 - \sigma_h, 1 + \sigma_h]$ respectively, where $\sigma_d \in [0, 1]$ and $\sigma_h \in [0, 1]$.*

Assumption 1.1 claims that although the overall demand trends are determined by selling prices, the demands fluctuate randomly. How the real demand varies is controlled by the distribution of a random variable. Assumption 1.2 states that the random demand is uniformly distributed among $[(1 - \sigma_d)D_d, (1 + \sigma_d)D_d]$ or $[(1 - \sigma_h)D_h, (1 + \sigma_h)D_h]$. Note that if $\sigma_d = \sigma_h = 1$, the demand will uniformly vary from 0 to $2D_d$ (or $2D_h$), which could contain the situation that there is no demand for the drug in the period.

**Assumption 2.** *$p_d$ and $p_h$ are defined on $[c, \overline{p}_d]$ and $[\phi c, \overline{p}_h]$. Here, $\overline{p}_d$ and $\overline{p}_h$ are the maximum selling price of drugstore and hospital, respectively.*

This assumption makes sense for China because the price of self-pricing pharmaceuticals is governed by the National Development and Reform Commission's regulations. This prevents the drug sellers from increasing the price without limitation. This assumption is also coincide with [16].

**Assumption 3.** *Assumptions on the demand function $D_d$ and $D_h$:*

1. $D_d$ is twice continuous differentiable in $p_d$ on $[c, \overline{p}_d]$, so is $D_h$ in $p_h$ on $[\phi c, \overline{p}_h]$.
2. $\frac{\partial D_d(p_d, p_h, A_d)}{\partial p_d} < 0$, $\frac{\partial D_h(p_d, p_h, A_h)}{\partial p_h} < 0$.
3. $\frac{\partial D_d(p_d, p_h, A_d)}{\partial p_h} > 0$, $\frac{\partial D_h(p_d, p_h, A_h)}{\partial p_d} > 0$.
4. $\frac{\partial D_d(p_d, p_h, A_d)}{\partial A_d} > 0$, $\frac{\partial D_h(p_d, p_h, A_h)}{\partial A_h} > 0$ and $A_h \geqslant A_d$.

Assumptions 3.2 and 3.3 indicate that the demand is decreasing with the seller's own price and increasing with his competitor's price. Assumption 3.4 means that the seller's demand will be higher when the seller is thought to be more reliable. As [12] described, for reasons such as physician recommendation, greater assurance of pharmaceutical quality, and convenience, patients prefer hospital pharmacies to drugstores.

**Assumption 4.** *Assumptions on the price elasticity of $e_d$ and $e_h$:*

1. $e_d = -\frac{\partial D_d/\partial p_d}{D_d/p_d}$ and $e_h = -\frac{\partial D_h/\partial p_h}{D_h/p_h}$ are increasing with $p_d$ and $p_h$, respectively. i.e., $\partial e_d/\partial p_d > 0$ and $\partial e_h/\partial p_h > 0$.
2. $e_d$ and $e_h$ are non-increasing with $p_h$ and $p_d$ respectively, i.e., $\partial e_d/\partial p_h \leq 0$ and $\partial e_h/\partial p_d \leq 0$.

Assumptions 4.1 and 4.2 tell us that the increase in one player's price will not only decrease expected demand but also reduce its competitor's price elasticity. This is an important property to show the supermodularity of game.

**Assumption 5.** *The domination condition: $\partial e_d/\partial p_d + \partial e_d/\partial p_h \geq 0$ and $\partial e_h/\partial p_h + \partial e_h/\partial p_d \geq 0$ are satisfied for the drugstore and hospital.*

Assumption 5 means that variation of the local price has a larger influence than that of its competitor. Later, we use this assumption to show the uniqueness of the Nash equilibrium point.

For the deterministic part of demand $D_d$ and $D_h$, there are two commonly used forms in the literature of single newsvendor with pricing decisions [16]:

1. The linear form:

$$D_d = A_d - a_d p_d + b_d p_h, \ D_h = A_h - a_h p_h + b_h p_d. \tag{1}$$

   where $A_d$, $A_h$, $a_d$, $a_h$, $b_d$, $b_h$ are all positive.
2. The logarithmic form:

$$D_d = \frac{A_d e^{-a_d p_d}}{A_d e^{-a_d p_d} + A_h e^{-a_h p_h}}, \ D_h = \frac{A_h e^{-a_h p_h}}{A_d e^{-a_d p_d} + A_h e^{-a_h p_h}}. \tag{2}$$

   where $A_d$, $A_h$, $a_d$, $a_h$ are all positive.

Here, the reliability factors of sellers depict the potential market demands. Next, we introduce a lemma to show that the commonly used demand functions are compatible with Assumptions 3 and 4. For simplicity, all proofs of theorems and lemmas can be found in Appendix A.

**Lemma 1.** *The commonly used demand functions given in (1) and (2) satisfy Assumptions 3 and 4.*

This lemma indicates that Assumptions 3 and 4 are reasonable in normal circumstances. Next section is devoted to the model formulation.

## 4. Model Formulation and Analysis

### 4.1. The Optimal Order Quantities of Hospital and Drugstore

As described in the previous section, the two sellers first decide their optimal order quantities and selling prices according to the market. Then, based on their decisions, the pharmaceutical supplier decides the two ex-factory prices to maximize total profit. The profit function of the drugstore can be written as:

$$\Pi_d(p_d, p_h, Q_d) = p_d E_{\xi_d}[min\{Q_d, D_d(p_d, p_h, A_d) \cdot \xi_d\}] - cQ_d. \tag{3}$$

The profit function of hospital can be written as:

$$\Pi_h(p_d, p_h, Q_h) = p_h E_{\xi_h}[min\{Q_h, D_h(p_d, p_h, A_h) \cdot \xi_h\}] - \phi cQ_h. \tag{4}$$

Given the selling price $p_h$ of the hospital, the drugstore faces a newsvendor pricing problem. We rewrite its profit function as $\Pi_d(p_d, p_h, Q_d) = p_d\left\{D_d \int_0^{Q_d/D_d} t f_{\xi_d}(t)dt + Q_d \int_{Q_d/D_d}^{\infty} f_{\xi_d}(t)dt\right\} - cQ_d$.

Note that $D_d$ has nothing to do with $Q_d$, so taking the partial derivative of $Q_d$, we obtain

$$\frac{\partial \Pi_d(p_d, p_h, Q_d)}{\partial Q_d} = p_d\left\{\frac{Q_d}{D_d}f_{\xi_d}(\frac{Q_d}{D_d}) + \int_{Q_d/D_d}^{\infty} f_{\xi_d}(t)dt - \frac{Q_d}{D_d}f_{\xi_d}(\frac{Q_d}{D_d})\right\} - c$$

$$= p_d \int_{Q_d/D_d}^{\infty} f_{\xi_d}(t)dt - c.$$

Further notice that $\frac{\partial^2 \Pi_d(p_d, p_h, Q_d)}{\partial (Q_d)^2} = -\frac{p_d}{D_d}f_{\xi_d}(\frac{Q_d}{D_d}) < 0$. So $\Pi_d$ is a concave function with respect to $Q_d$.

Set $\frac{\partial \Pi_d(p_d, p_h, Q_d)}{\partial Q_d} = 0$, we could see that the optimal order quantity $Q_d^*$ is

$$Q_d^* = D_d(p_d, p_h, A_d) \cdot F_{\xi_d}^{-1}(1 - \frac{c}{p_d}). \tag{5}$$

Repeating the same procedure again, we know that $\Pi_h$ is also concave with respect to $Q_h$ and the optimal order quantity of the hospital is

$$Q_h^* = D_h(p_d, p_h, A_h) \cdot F_{\xi_h}^{-1}(1 - \frac{\phi c}{p_h}). \tag{6}$$

Next, we claim that both of $\Pi_d$ and $\Pi_h$ are concave with respect to $p_d$ and $p_h$, respectively.

**Lemma 2.** *Both of the profit functions $\Pi_d(p_d, p_h, Q_d)$ and $\Pi_h(p_d, p_h, Q_h)$ are concave with respect to $p_d$ and $p_h$, respectively.*

Lemma 2 and the concavity of $\Pi_d$ (with respect to $Q_d$) ensure that Equation (5) will always hold in this game, for if it does not hold, the decision-maker of the drugstore could change its order quantity to $Q_d^*$ to gain a higher profit. So given $p_h$ of the hospital, the only option for the drugstore is to choose an optimal $p_d$ to maximize its profit. This is also true for the hospital.

Substituting $Q_d$ with the optimal $Q_d^*$ in Equation (3), the response function of the drugstore is $\Pi_d(p_d, p_h, Q_d^*) = D_d p_d \int_0^{F_{\xi_d}^{-1}(\rho_d)} t f_{\xi_d}(t)dt$, where $\rho_d = 1 - c/p_d$.

Similarly, we have $\Pi_h(p_d, p_h, Q_h^*) = D_h p_h \int_0^{F_{\xi_h}^{-1}(\rho_h)} t f_{\xi_h}(t) dt$, where $\rho_h = 1 - \phi c / p_h$. By Assumption 12, $\xi_d$ is uniformly distributed on $[1 - \sigma_d, 1 + \sigma_d]$.

So $f_{\xi_d}(t) = \begin{cases} 1 & t \in [1 - \sigma_d, 1 + \sigma_d] \\ 0 & \text{elsewhere} \end{cases}$ , $F_{\xi_d}(t) = \begin{cases} 0 & t \in [0, 1 - \sigma_d) \\ \frac{t - (1 - \sigma_d)}{2\sigma_d} & t \in [1 - \sigma_d, 1 + \sigma_d] \\ 1 & t \in (1 + \sigma_d, \infty) \end{cases}$ .

Thus we have $\Pi_d(p_d, p_h, Q_d^*) = 4 D_d p_d \sigma_d (1 - \frac{\sigma_d c}{p_d})(1 - \frac{c}{p_d})$ and $\Pi_h(p_d, p_h, Q_h^*) = 4 D_h p_h \sigma_h (1 - \frac{\sigma_h \phi c}{p_h})(1 - \frac{\phi c}{p_h})$.

Next section is devoted to the equilibrium analysis.

### 4.2. Equilibrium Analysis

Equilibrium is achieved in a system when all of the relating factors compete on a balanced level. It is common sense that in a pricing game, the players' prices will reach certain points during a long period of competition. Thus, analyzing the properties of equilibrium points is very important. This section considers the existence and uniqueness of the Nash equilibrium in this game.

#### 4.2.1. Existence of Nash Equilibrium

By the definition [61], the Nash equilibrium point is a $(p_d^*, p_h^*) \in [c, \overline{p}_d] \times [\phi c, \overline{p}_h]$ that satisfies $\Pi_d(p_d^*, p_h^*, Q_d^*) \geqslant \Pi_d(p_d, p_h^*, Q_d^*)$ and $\Pi_h(p_d^*, p_h^*, Q_h^*) \geqslant \Pi_h(p_d^*, p_h, Q_h^*)$ for all $(p_d, p_h) \in [c, \overline{p}_d] \times [\phi c, \overline{p}_h]$.

In order to prove the existence, we first prove a lemma.

**Lemma 3.** *The game under consideration is supermodular.*

Lemma 3 actually guarantees the existence of the Nash Equilibrium. Furthermore, ref. [62] proved that there exists some simple algorithms to find an equilibrium point.

Now we discuss the existence of the Nash Equilibrium.

**Theorem 1.** *There exists at least one Nash Equilibrium in this game.*

The theorem above tells us that in this game, the hospital and the drugstore will reach an equilibrium under which no player has anything to gain by changing only their own strategy. In the next subsection, we discuss the uniqueness of the equilibrium.

#### 4.2.2. Uniqueness of Nash Equilibrium

Taking derivatives on $\Pi_d$ and $\Pi_h$ with respect to $p_d$ and $p_h$ separately, we obtain

$$\frac{\partial \Pi_d(p_d, p_h, Q_d^*)}{\partial p_d} = D_d \left[ (-e_d + 1) \cdot \int_0^{F_{\xi_d}^{-1}(\rho_d)} t f_{\xi_d}(t) dt + \frac{c}{p_d} \cdot F_{\xi_d}^{-1}(\rho_d) \right], \qquad (7)$$

$$\frac{\partial \Pi_h(p_d, p_h, Q_h^*)}{\partial p_h} = D_h \left[ (-e_h + 1) \cdot \int_0^{F_{\xi_h}^{-1}(\rho_h)} t f_{\xi_h}(t) dt + \frac{\phi c}{p_h} \cdot F_{\xi_h}^{-1}(\rho_h) \right]. \qquad (8)$$

All of the derivations of the equations in this paper are provided in Appendix B.

Let $\rho_d^*, \rho_h^*, e_d^*, e_h^*, D_d^*$ and $D_h^*$ denote $1 - c/p_d^*$, $1 - \phi c/p_h^*$, $e_d(p_d^*, p_h^*, A_d)$, $e_h(p_d^*, p_h^*, A_h)$, $D_d(p_d^*, p_h^*, A_d)$, and $D_h(p_d^*, p_h^*, A_h)$, respectively.

As the equilibrium point must satisfy $\begin{cases} \frac{\partial \Pi_d(p_d,p_h^*,Q_d^*)}{\partial p_d}\Big|_{(p_d,p_h)=(p_d^*,p_h^*)} = 0 \\ \frac{\partial \Pi_h(p_d^*,p_h,Q_h^*)}{\partial p_h}\Big|_{(p_d,p_h)=(p_d^*,p_h^*)} = 0 \end{cases}$. It is equiv-

alent to say that the following two equations: $-e_d^* + 1 + \frac{c}{p_d^*} \cdot \frac{F_{\xi_d}^{-1}(\rho_d^*)}{\int_0^{F_{\xi_d}^{-1}(\rho_d^*)} t f_{\xi_d}(t)dt} = 0$ and

$-e_h^* + 1 + \frac{\phi c}{p_h^*} \cdot \frac{F_{\xi_h}^{-1}(\rho_h^*)}{\int_0^{F_{\xi_h}^{-1}(\rho_h^*)} t f_{\xi_h}(t)dt} = 0$ must be satisfied simultaneously.

Recall that $\xi_d$ and $\xi_h$ are uniformly distributed on $[1-\sigma_d, 1+\sigma_d]$ and $[1-\sigma_h, 1+\sigma_h]$, respectively, we can rewrite the two equations above as $-e_d^* + 1 + \frac{2c}{p_d^*} \cdot \frac{-2\sigma_d c/p_d^*+1+\sigma_d}{(-2\sigma_d c/p_d^*+1+\sigma_d)^2-(1-\sigma_d)^2} = 0$ and $-e_h^* + 1 + \frac{2\phi c}{p_h^*} \cdot \frac{-2\sigma_h \phi c/p_h^*+1+\sigma_h}{(-2\sigma_h \phi c/p_h^*+1+\sigma_h)^2-(1-\sigma_h)^2} = 0$.

Define $g_d(p_d) \triangleq -e_d + 1 + \frac{2c}{p_d} \cdot \frac{-2\sigma_d c/p_d+1+\sigma_d}{(-2\sigma_d c/p_d+1+\sigma_d)^2-(1-\sigma_d)^2}$ and $g_h(p_h) \triangleq -e_h + 1 + \frac{2\phi c}{p_h} \cdot \frac{-2\sigma_h \phi c/p_h+1+\sigma_h}{(-2\sigma_h \phi c/p_h+1+\sigma_h)^2-(1-\sigma_h)^2}$, the Nash equilibrium point $(p_d^*, p_h^*)$ of the game under our consideration can be solved by condition

$$\begin{cases} g_d(p_d^*, e_d^*) = 0 \\ g_h(p_h^*, e_h^*) = 0 \end{cases}. \tag{9}$$

To show the quasi-concavity of the payoff functions, we establish the following two lemmas:

**Lemma 4.** $g_d(p_d)$ *is non-decreasing with $p_h$, and strictly monotonic decreasing with $p_d$; $g_h(p_h)$ is non-decreasing with $p_d$, and strictly monotonic decreasing with $p_h$. So the best response functions $r_d^*(p_h)$ and $r_h^*(p_d)$ can be uniquely determined by solving $g_d(p_d) = 0$ and $g_h(p_h) = 0$, respectively.*

**Lemma 5.** *Suppose $f : X \to \mathbb{R}$ is a twice continuously differentiable function defined on $X$, if $X \subset \mathbb{R}$, the f is quasi-concave if and only if it is monotonic or first non-decreasing and then non-increasing.*

**Lemma 6.** *The payoff functions $\Pi_d(p_d, p_h, Q_d^*)$ and $\Pi_h(p_d, p_h, Q_h^*)$ are quasi-concave in $p_d$ and $p_h$, respectively.*

Note that the equilibrium point $(p_d^*, p_h^*)$ must satisfy the first order conditions $g_d(p_d^*, e_d^*) = 0$ and $g_h(p_h^*, e_h^*) = 0$ simultaneously, the uniqueness of Nash equilibrium can be proved.

To obtain Theorem 2, we use the index theory approach [61], which is based on the Poincare-Hopf index theorem in differential topology [62].

**Theorem 2.** *In the game discussed, there exists a unique Nash equilibrium. It can be solved by Equation (9).*

Therefore, the hospital and the drugstore will reach and only reach a Nash equilibrium point in the single drug-selling competition. This proposition can be used for analyzing the causes of drug shortage problems with commonly used demand functions. In the next section, we discuss the optimal strategies for the hospital and drugstore with linear demand functions.

### 4.3. Pricing Analysis with Linear Demand Functions

Recall that in (1) that the linear form of demand functions are $D_d = A_d - a_d p_d + b_h p_h$ and $D_h = A_h - a_h p_h + b_d p_d$. We further assume $a_d a_h > \frac{1}{4} b_d b_h$, which implies that a change in the local price has a relatively larger influence on demand than that of non-local price. As we state in the comments about Assumption 1.2, to take the situation that nobody needs a drug in the period into consideration, we let $\sigma_d = \sigma_h = 1$. The real demand $RD_d$ and $RD_h$ will then vary uniformly between $[0, 2D_d]$ and $[0, 2D_h]$.

Now we have $e_d^* = \frac{a_d p_d^*}{A_d - a_d p_d^* + b_d p_h}$ and $e_h^* = \frac{a_d p_h^*}{A_h - a p_h^* + b p_d}$. Substituting $\sigma_d = \sigma_h = 1$ into condition (9) we know that at the equilibrium point, $p_d^*$ and $p_d^*$ must satisfy

$$\begin{cases} -\frac{a_d p_d^*}{A_d - a_d p_d^* + b_d p_h^*} + 1 + \frac{2c}{p_d^*} \cdot \frac{1}{(-2c/p_d^* + 2)} = 0 \\ -\frac{a_h p_h^*}{A_h - a_h p_h^* + b_h p_d^*} + 1 + \frac{2\phi c}{p_h^*} \cdot \frac{1}{(-2\phi c/p_h^* + 2)} = 0 \end{cases}.$$

After some simplifications, we have $\begin{cases} p_d^* = \frac{a_d c + b_d p_h^* + A_d}{2a_d} \\ p_h^* = \frac{a_h \phi c + b_h p_d^* + A_h}{2a_h} \end{cases}.$

Solve the equations, we obtain

$$\begin{cases} p_d^* = \frac{2a_d a_h c + a_h b_d \phi c + 2a_h A_d + b_d A_h}{4a_d a_h - b_d b_h} \\ p_h^* = \frac{2a_d a_h \phi c + a_d b_h c + 2a_d A_h + b_h A_d}{4a_d a_h - b_d b_h} \end{cases}. \tag{10}$$

We can see in Equation (10) that $p_d^*$ and $p_h^*$ are increasing with $c$ and $\phi$. It is intuitive that the increasing ex-factory price will cause retailers' selling prices to increase. The decrease in discounts for the hospital is very similar to an increase in its marginal cost, thus increasing its optimal price. An increase in the hospital's marginal cost will have a positive effect on the drugstore's demand, which will increase the drugstore's optimal price.

At the equilibrium point, for the drugstore and hospital, we have $Q_d^* = D_d(p_d^*, p_h^*, r_d) \cdot F_{\xi_d}^{-1}(1 - \frac{c}{p_d^*}) = (A_d - a_d p_d^* + b_d p_h^*) \cdot (-\frac{2c}{p_d^*} + 2)$ and $Q_h^* = D_h(p_d^*, p_h^*, r_h) \cdot F_{\xi_h}^{-1}(1 - \frac{\phi c}{p_h^*}) = (A_h - a_h p_h^* + b_h p_d^*) \cdot (-\frac{2\phi c}{p_h^*} + 2)$.

We define $s_d^* \triangleq P\{\xi_d | Q_d^* > RD_d^*\}$ and $s_h^* \triangleq P\{\xi_h | Q_h^* > RD_h^*\}$ as the satisfaction rates of the drugstore and hospital, respectively. Note that as $RD_d^* = D_d^* \cdot \xi_d$, $RD_h^* = D_h^* \cdot \xi_h$, and $\xi_d$, $\xi_h$ are uniformly distributed on $[0, 2]$, we have $s_d^* = Q_d^*/2D_d^*$ and $s_h^* = Q_h^*/2D_h^*$. To explain the drug shortage problem, we need the following theorem:

**Theorem 3.** *In the linear demand case, the satisfaction rates of the two sellers have the following properties:*

1.  $s_d^*$ *is monotonically decreasing with $c$ and increasing with $\phi$.*
2.  $s_h^*$ *is monotonically decreasing with $c$ and $\phi$.*

Theorem 3 depicts some possible reasons for a drug shortage problem. If for some reason, such as a lack of raw materials or the COVID-19 pandemic, the pharmaceutical supplier increases the ex-factory price of a certain drug, it will reduce the drugstore's and hospital's satisfaction rates, which could cause a drug shortage problem. Government policy also plays a role in drug shortages. If the government stops the hospital's discounts, it will be more difficult for the hospital to retain a high stock, leading to a drug shortage in the hospital. When the hospital loses some power in the game, the drugstore's market share may be increased, which is why we see $s_d^*$ increases with $\phi$ in Theorem 3.1.

As discussed in Section 1, it is common for drugs to be cheaper in a drugstore than in a hospital. Thus, when people suffer common or chronic diseases, they often prefer drugstores to hospitals. In the next two subsections, we describe two special kinds of linear demand to show that in some cases, the game ends up with $p_d^* > p_h^*$, as reported in [15].

4.3.1. Symmetric Linear Demand

Let $a_d = a_h = l$ represent the impact of local price change. Further let $b_d = b_h = k$ represent the impact of cross-price change. The demand function will then change to $D_d = A_d - l p_d + k p_h$ and $D_h = A_h - l p_h + k p_d$, respectively.

We call it symmetric because the demand functions above will lead to a symmetrical position of two sellers. Both sellers will face the same local price change impact rate. That is, the increase of one player's selling price will cause a decrease of its demand at a certain rate (say $l/yuan$), and the increase of its competitor's selling price will cause an increase of its

demand at a certain rate (say $k/yuan$). The price change impact factors can be regarded as a measurement of the market sensitivity: the higher they are, the more sensitive market is.

Based on this consideration, we have

$$\begin{cases} p_d^* = \frac{2l^2c + lk\phi c + 2lA_d + kA_h}{4l^2 - k^2} \\ p_h^* = \frac{2l^2\phi c + lkc + kA_d + 2lA_h}{4l^2 - k^2} \end{cases}. \tag{11}$$

To consider the influence of local price change impact $l$ and cross-price change impact $k$, we establish the following theorem:

**Theorem 4.** *In the symmetric linear demand case, the optimal prices and satisfaction rates of the two sellers: $p_d^*$, $p_h^*$, $s_d^*$ and $s_h^*$ are all monotonically decreasing with $l$ and increasing with $k$.*

Theorem 4 states that in the symmetric case, when the market is relatively sensitive to its own price, the seller will have a lower optimal price and satisfaction rate. When the market is relatively sensitive to its competitor's price, the seller will have a higher optimal price and satisfaction rate. According to our model, the local price change impact for each player has a negative effect on demand, whereas the cross-price change impact has a positive effect. Thus, the factor $l/k$ can be interpreted as an indicator of market sensitivity. A larger $l$ or smaller $k$ represents a highly sensitive market, in which it is difficult for the players to hold high stocks and prices. In contrast, a less sensitive market allows the players to hold more stock and set high prices. Thus, a market with high sensitivity could be another cause of drug shortage problems.

Comparing the two prices at the equilibrium point, we obtain $p_h^* - p_d^* = \frac{(A_h - A_d) - l(1 - \phi)c}{2l + k}$.

This tells us that there are two groups of influential factors whose price would be higher. One is affected by the markets, and it is depicted by the ex-factory price, local price change impact, and discount factor; the other is determined by patients, and it can be characterized by sellers' reliability or patients' perception of a curative effect.

If $A_h - A_d \geq l(1 - \phi)c$, it will lead to $p_d^* \leq p_h^*$ at the equilibrium point. That is, compared to the difference in reliability between the hospital and drugstore, if the price gap between them is not very large and the price does not strongly influence demand, the drugstore will not have a higher selling price than the hospital. A special case of this situation is $\phi = 1$, which means that the government gives no discount to the hospital. As the hospital has a higher reliability and thus a higher potential demand, it could increase its price to gain more profits.

In a less sensitive market, when the hospital has a lower marginal cost than the drugstore, and patients' perceptions of the difference in reliability or curative effect between the drugstore and hospital is small enough (smaller than $l(1 - \phi)c$), we may have $p_h^* < p_d^*$ at the equilibrium point.

### 4.3.2. Seller-Reliant Linear Demand

Let $a_d = b_h = d$ represent the price change impact of the drugstore. Further let $a_h = b_d = h$ represent the price change impact of the hospital. The demand function will then change to $D_d = A_d - dp_d + hp_h$ and $D_h = A_h - hp_h + dp_d$, respectively.

We call this situation seller-reliant demand because the impact of price change is determined by who sells the drug. Changing the hospital's selling price will always cause a demand impact of $h/yuan$ and changing the drugstore's selling price will cause an impact of $d/yuan$. Under this consideration, a one-unit increase in the hospital's (or drugstore's) selling price would decrease its own demand by $h$ (or $d$) units and increases its competitor's demand by $d$ (or $h$) units.

In the seller-reliant linear demand case, the price change impact actually depicts how far the drugstore and hospital can influence the market. This game is no longer symmetric as we use $d$ and $h$ to differentiate the two sellers, respectively.

Similar to Section 4.3.1, we could work out

$$\begin{cases} p_d^* = \frac{2dc + h\phi c + 2A_d + A_h}{3d} \\ p_h^* = \frac{dc + 2h\phi c + A_d + 2A_h}{3h} \end{cases}. \tag{12}$$

A theorem is also established to consider the influence of two sellers' price change impact:

**Theorem 5.** *In the linear demand case, the optimal prices and satisfaction rates of the two sellers have the following properties:*

1. $p_d^*$ *is monotonically decreasing with d and increasing with h, $p_h^*$ is monotonically decreasing with h and increasing with d.*
2. $s_d^*$ *is monotonically decreasing with d and increasing with h, $s_h^*$ is monotonically decreasing with h and increasing with d.*

Note $d$ and $h$ describe the price sensitivity of the drugstore and hospital, respectively. It is natural that for each player, the optimal price will decrease with its own price sensitivity and increase with its competitor's price sensitivity. The satisfaction rates follow the same pattern because a player with high price sensitivity is less likely to satisfy the demand, which leads to a low satisfaction rate. Thus, high price sensitivity could be one of the reasons for a drug shortage problem experienced by one particular retailer.

Comparing the two optimal prices, we have

$$p_h^* - p_d^* = \frac{(d - 2h)(dc + A_d) + (2d - h)(h\phi c + A_h)}{3dh}. \tag{13}$$

To show when situation $p_h^* < p_d^*$ will happen, it is necessary to consider the value of two price change impacts, $d$ and $h$.

1. If $d \geqslant 2h$, that is, in the market, the drugstore has an influence at least twice that of the hospital. We obtain that $(d - 2h)(dc + A_d)$ is non-negative and $(2d - h)(h\phi c + A_h)$ is positive. Thus in this situation, the game will end up with $p_h^* > p_d^*$.
2. If $d \leqslant \frac{h}{2}$, the drugstore has an influence of half or less than half that of the hospital. Under this circumstance. We obtain that $(d - 2h)(dc + A_d)$ is non-positive and $(2d - h)(h\phi c + A_h)$ is negative. Thus, at the equilibrium point, the hospital will have a lower selling price than the drugstore.
3. If $\frac{h}{2} < d < 2h$, we rewrite Equation (13) as

$$p_h^* - p_d^* = \frac{cd^2 + (A_d + 2A_h + 2h\phi c - 2hc)d - h^2\phi c - hA_h - 2hA_d}{3dh}.$$

Define $u(d) \triangleq cd^2 + (A_d + 2A_h + 2h\phi c - 2hc)d - h^2\phi c - hA_h - 2hA_d$. Investigate the sign of Equation (13) is then equivalent to discuss the sign of $u(d)$ on $(\frac{h}{2}, 2h)$.

Taking the derivative of $u(d)$ on $d$, we have $u'(d) = 2cd + A_d + 2A_h + 2h\phi c - 2hc$.

So the minimum point of $h(a)$ is $d^* = h - \frac{A_d + 2A_h + 2h\phi c}{2c}$.

As $d^* < h$, we know by the symmetry of $h(d)$ that $u(d^*) \leqslant u(\frac{h}{2}) < u(2h)$. In case 1 and 2, we have already shown that $u(\frac{h}{2}) < 0$ and $u(2h) > 0$, there must exist a unique $d_0$ such that $u(d_0) = 0$. We could obtain $d_0 = \frac{\sqrt{K^2 + 4cL} - K}{2c}$ by solving the equation $u(d) = 0$, where $K = A_d + 2A_h + 2h\phi c - 2hc$ and $L = h^2\phi c + hA_h + 2hA_d$. The expression of $d_0$ is not important, what we are concerned about is that when $\frac{h}{2} < d < d_0$, the game will have $p_d^* > p_h^*$ at the equilibrium point, and when $d_0 \leqslant d < 2h$, it will end up with $p_d^* \leqslant p_h^*$.

Now we can see that $p_d^* > p_h^*$ is possible in seller-reliant demand. That is, when the price change impact of the drugstore, $d$ is small enough (smaller than $d_0$), the hospital will choose a selling price even lower than that of the drugstore. When the influence of the drugstore's price is relatively weak, increasing it will not cause a big reduction in demand,

so the drugstore will increase its selling price to gain more profit, subsequently leading to $p_d^* > p_h^*$ at the equilibrium point.

We now summarize the discussions above. A drug shortage problem may have three causes. The first is the high price sensitivity of the market, which makes it difficult for retailers to keep enough stock. Second, an increase in ex-factory price will cost the business more to satisfy demand. Third, if the government offers smaller discounts to a hospital, it will cause its satisfaction rate to reduce. The price difference between the drugstore and hospital can be divided into two cases. In the symmetric linear demand case, both sellers face a symmetric price change impact; the only advantage of the hospital is reflected in the difference between two reliability or curative effects, $A_h$ and $A_d$. If $A_h - A_d$ is relatively small, a highly sensitive market with a large discount for the hospital may result in the uncommon situation of $p_d^* > p_h^*$. In the seller-reliant demand case, parameter $d$ is used to depict the price change impact of the drugstore and $h$ characterizes the price change impact of the hospital. If $d$ is small enough, the drugstore can increase its price to gain more profit, which may lead the hospital to have a relatively lower selling price. These results explain the price disparity in [15].

To further illustrate the factors contributing to the drug shortage and the price disparity problems, we present numerical studies in the next section.

## 5. Numerical Analysis

In this section, we analyze the decisions of the drugstore and hospital in both symmetric and seller-reliant cases. For each case, we investigate the influence of the price change impact, ex-factory price, and discount factor on each player's optimal prices, order quantities, expected profits, and satisfaction rates.

### 5.1. Symmetric Linear Demand

As shown in Section 4.3.1, when the cross price change impact $k$ is fixed, the local price change impact $l$ can be regarded as a indicator of market sensitivity. To consider the influence of local price change impact $l$, ex-factory price $c$ and discount factor $\phi$, we set $A_d = 1000$, $A_h = 1100$ and draw four groups of functions.

Figure 3 shows that the two lines in all three sub-figures intersect with each other, which means that the drugstore can have a higher optimal price than the hospital. It is also clear that the optimal prices of the two retailers will decrease with price sensitivity and discounts provided for the hospital and increase with the ex-factory price. This is intuitive because large price sensitivity, a low ex-factory price, and small discounts often lead to a lower optimal price. Figure 3a shows that with the advantages of purchasing discount and higher reliability, the hospital could set a higher optimal price than the drugstore but set a lower one with a relatively large $l$. When the market becomes more sensitive, the two players may tend to use a so-called small profit and quick turnover policy. Thus, the hospital could offer a lower price to gain market share. Figure 3b,c show two other possible situations of $p_d^* > p_h^*$. The first occurs because when the ex-factory price is relatively small, the advantage of higher reliability allows the hospital to select a higher optimal price. When the ex-factory price increases, the discount provided by the government gives the hospital a much lower marginal cost, which leads to a result of $p_h^* < p_d^*$. The second situation occurs because when the discount provided to the hospital is very large, it has a cost advantage and thus charges a lower price.

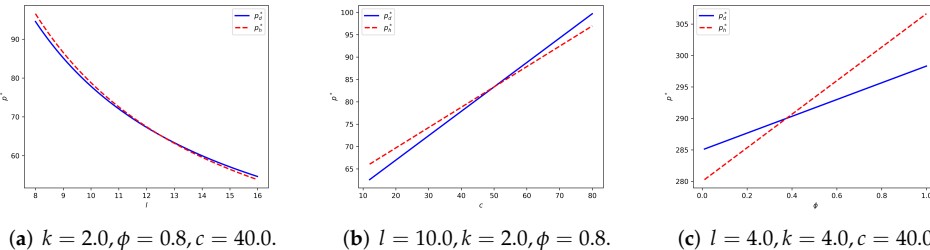

(**a**) $k = 2.0, \phi = 0.8, c = 40.0$.     (**b**) $l = 10.0, k = 2.0, \phi = 0.8$.     (**c**) $l = 4.0, k = 4.0, c = 40.0$.

**Figure 3.** Optimal Prices of the Symmetric Demand Case.

Figure 4 illustrates the influence of the local price change impact, ex-factory price, and discount factor on the optimal order quantities. As the demands are symmetric, the two lines in the three sub-figures do not intersect and the hospital always has a higher optimal order quantity due to its discount and reliability advantages. It can be seen in Figure 4a,b that the optimal order quantities for both players decrease with the local price change impact and ex-factory price because either a higher market sensitivity or ex-factory price will make the retailers order less. Figure 4c shows that if the government reduces discounts for the hospital, the optimal order quantity of the hospital will be reduced but that of the drugstore will be slightly increased. When the hospital's marginal cost increases, it will order less and therefore the market share of its competitor will increase.

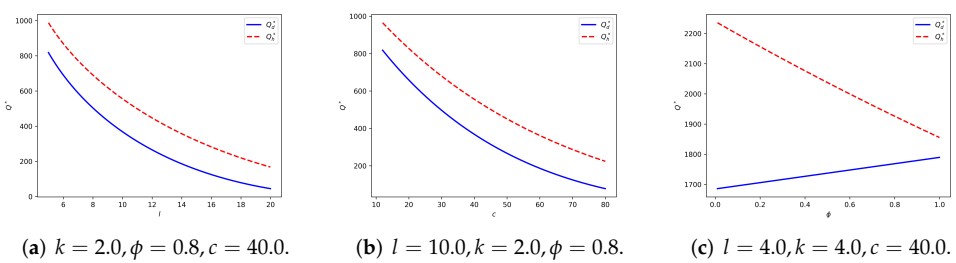

(**a**) $k = 2.0, \phi = 0.8, c = 40.0$.     (**b**) $l = 10.0, k = 2.0, \phi = 0.8$.     (**c**) $l = 4.0, k = 4.0, c = 40.0$.

**Figure 4.** Optimal Order Quantities of the Symmetric Demand Case.

Figure 5 shows that the expected profits of the two players follow almost the same pattern as the optimal quantities. The expected profits also decrease with the local price change impact and ex-factory price, where the decrease with the local price change impact is even steeper (see Figure 5a,b). Figure 5c depicts the influence of the discount factor: when the government offers smaller discounts for the hospital, its expected profit is lower but the drugstore gains more profits. As the hospital has the advantages of reliability and $\phi \in (0, 1]$, it will always gain more profit than the drugstore regardless of the values of the local price change impact, ex-factory price, and discount factor.

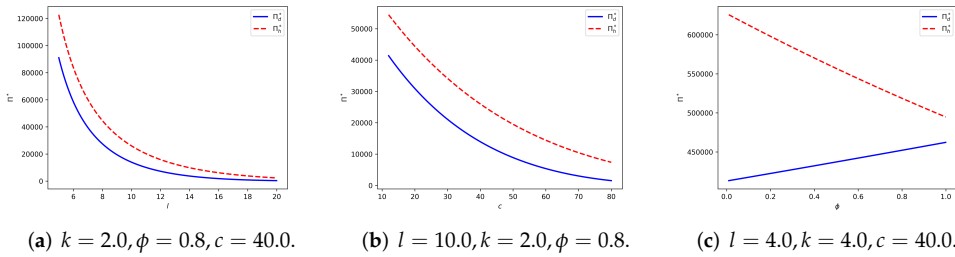

(**a**) $k = 2.0, \phi = 0.8, c = 40.0$.     (**b**) $l = 10.0, k = 2.0, \phi = 0.8$.     (**c**) $l = 4.0, k = 4.0, c = 40.0$.

**Figure 5.** Expected Profits of the Symmetric Demand Case.

To gain insight into the drug shortage problem, we also need to investigate the influence of these factors on the satisfaction rate. As stated in Theorems 3 and 4, Figure 6a,b show two possible causes of the drug shortage problem. One is a highly sensitive market.

Neither the hospital nor the drugstore can handle highly price-sensitive demand with uncertainty. The other is an ex-factory price that is too high for the two players to hold enough stock to manage uncertain demand. In conjunction with Theorem 3, Figure 6c shows that decreasing the government discount reduces the satisfaction rate of the hospital and increases that of the drugstore. The increasing marginal cost may make it more difficult for the hospital to retain stock. As a consequence, the drugstore can gain some power in the game and thus it slightly increases its satisfaction rate.

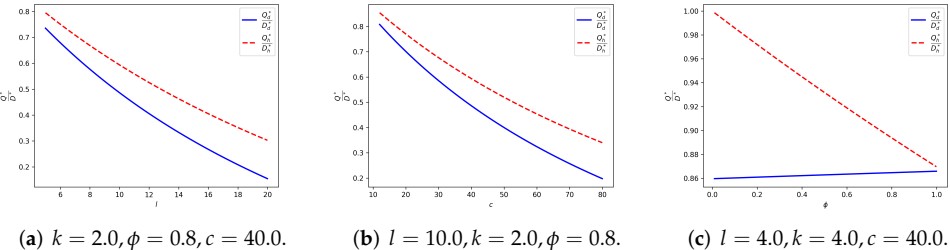

(**a**) $k = 2.0, \phi = 0.8, c = 40.0$.　　(**b**) $l = 10.0, k = 2.0, \phi = 0.8$.　　(**c**) $l = 4.0, k = 4.0, c = 40.0$.

**Figure 6.** Satisfaction Rate of the Symmetric Demand Case.

The next subsection is devoted to the seller-reliant demand case, in which there is another kind of competition.

### 5.2. Seller-Reliant Linear Demand

Unlike the symmetric case, under the seller-reliant linear demand function, price sensitivity differs not according to local or non-local factors but to the participants in the competition. When $h$ is fixed, the drugstore's price change impact $d$ can be regarded as another indicator of market sensitivity. If $d$ is comparatively small relative to $h$, the drugstore customers are less price sensitive than those of the hospital, and vice versa. Let $A_d = 1000$, $A_h = 1200$. As in the former subsection, four groups of graphs are drawn to illustrate the factors contributing to optimal prices, quantities, profits, and satisfaction rate.

Figure 7 shows that in the seller-reliant linear demand case, it is possible for the drugstore to have a higher optimal price than the hospital. In Figure 7a, it is intuitive to see that when the price sensitivity of the drugstore increases, the optimal price of the hospital will increase and that of the drugstore will decrease as the price change impact of the hospital is fixed. Thus, there exists a threshold of the drugstore's price change impact below which drugs will be cheaper in the drugstore than in the hospital. Figure 7b,c show that the optimal prices of the two retailers increase with the ex-factory price and decrease with the discounts provided to the hospital. Similar to the symmetric demand case, the game may end with $p_d^* > p_h^*$ when the ex-factory price is high or the discount for the hospital is small.

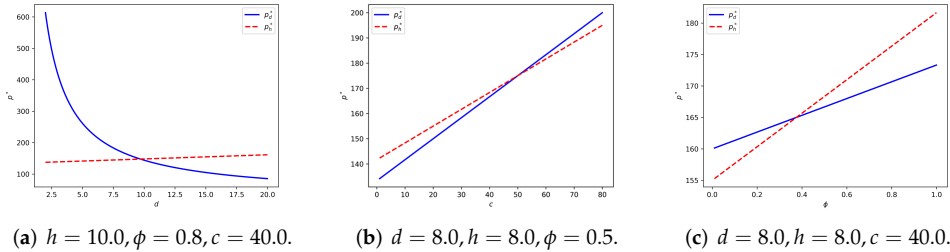

(**a**) $h = 10.0, \phi = 0.8, c = 40.0$.　　(**b**) $d = 8.0, h = 8.0, \phi = 0.5$.　　(**c**) $d = 8.0, h = 8.0, c = 40.0$.

**Figure 7.** Optimal Prices of the Seller-relied Demand Case.

Figure 8 depicts the factors contributing to optimal order quantities. Figure 8a indicates that a seller with comparatively higher price sensitivity may have a lower optimal order quantity. Instead of $d = 10.0$, the threshold where $Q_h^*$ exceeds $Q_d^*$ occurs on $d < 10.0$. The reason is that the advantages of discounts and reliability make the hospital more competitive. It is intuitive that the hospital's optimal order quantity decreases with the ex-factory price and increases with the government discount and that the drugstore's

optimal order quantity decreases with them (see Figure 8b,c). When the drugstore has a distinct advantage in price sensitivity, there may exist thresholds for the ex-factory price and discount factor, above which the drugstore will order more than the hospital.

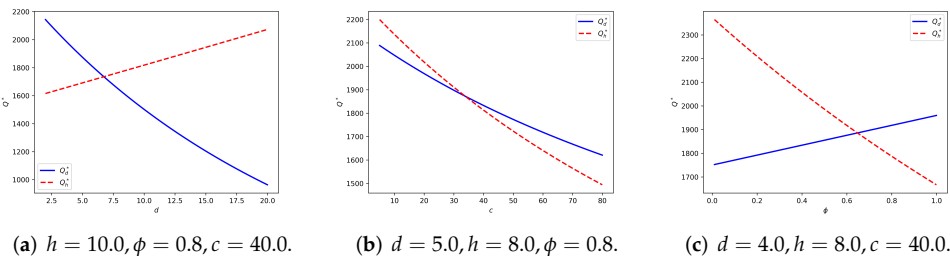

(**a**) $h = 10.0, \phi = 0.8, c = 40.0.$     (**b**) $d = 5.0, h = 8.0, \phi = 0.8.$     (**c**) $d = 4.0, h = 8.0, c = 40.0.$

**Figure 8.** Optimal Quantities of the Seller-relied Demand Case.

The optimal expected profits of the two retailers follow a similar pattern to the order quantities. Figure 9a shows that if the drugstore has a distinct advantage in price sensitivity, it has a larger profit than the hospital. Figure 9b shows that increasing the ex-factory price will increase the marginal cost of both players, leading to a decrease in their expected profits. When the market is less sensitive to the drugstore's price and the ex-factory price is relatively low, the drugstore may have a larger profit than the hospital. As the ex-factory price increases, the hospital's profit gradually increases via its purchasing discounts. It can be seen in Figure 9c that if the government reduces the hospital's discounts, the profit of the hospital will decrease and that of the drugstore will increase. The reduction in the discount reduces the hospital's profits, giving the drugstore an opportunity to increase its market share.

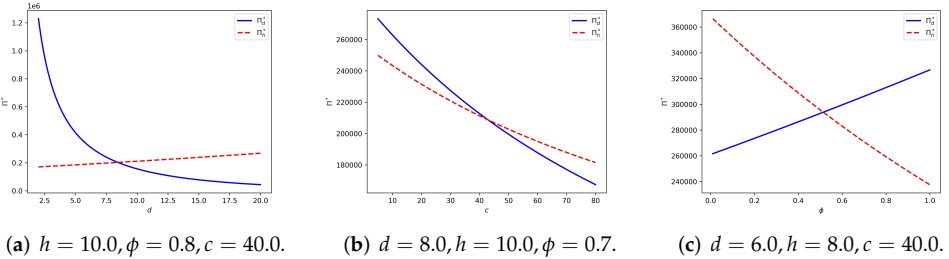

(**a**) $h = 10.0, \phi = 0.8, c = 40.0.$     (**b**) $d = 8.0, h = 10.0, \phi = 0.7.$     (**c**) $d = 6.0, h = 8.0, c = 40.0.$

**Figure 9.** Optimal Expected Profits of the Seller-relied Demand Case.

Figure 10 depicts the satisfaction rates for the three contributing factors. Figure 10a indicates that for each player, the satisfaction rate will decrease with its own price change impact. That is, if only one of the two sellers faces the shortage problem, it may be caused by the high price sensitivity of this particular seller. A high ex-factory price may be another reason for the drug shortage problem (see Figure 10b) because an increase in the ex-factory price will increase the cost of holding stock. Figure 10c shows that the government discount also influences the hospital's drug shortage: if a larger discount is given to the hospital, it is less likely that the drug will go out of stock.

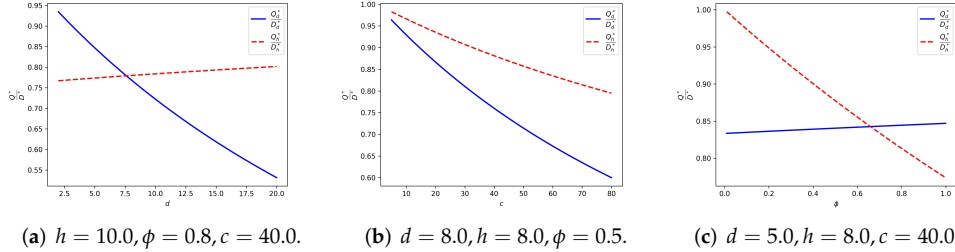

(**a**) $h = 10.0, \phi = 0.8, c = 40.0.$     (**b**) $d = 8.0, h = 8.0, \phi = 0.5.$     (**c**) $d = 5.0, h = 8.0, c = 40.0.$

**Figure 10.** Satisfaction Rate of the Seller-relied Demand Case.

## 6. Conclusions

Motivated by the current state of China's pharmaceutical supply chain, we consider the drug shortage and price disparity problems in the country. This paper establishes a model with one pharmaceutical supplier, one hospital, and one drugstore in a decentralized supply chain with price-sensitive and uncertain demand. By analyzing a game between the retailers, we find that our pharmaceutical supply chain game has a unique Nash equilibrium point. The closed-form solution of optimal strategies can be found in linear demand cases.

The three factors contributing to the drug shortage problem are the ex-factory price, the price sensitivity of the market, and the government discount provided to the hospital. High ex-factory price and price sensitivity of the market or small discounts cut the retailers' profits, thus causing them to stop ordering.

We also show that in most cases of drug price disparity, hospitals can set a higher optimal selling price than drugstores due to their reliability and discount advantages. However, there are exceptions in daily life [15]. We propose two special kinds of linear demand to provide some insights into this unusual phenomenon. In the symmetric demand case, if the drugstore is almost as reliable as the hospital, high local price sensitivity, high discounts, or a high ex-factory price may cause the drugstore to set a higher price than the hospital. In the seller-reliant demand case, low drugstore price sensitivity, a large discount for the hospital, or a high ex-factory price may also result in atypical phenomena.

### 6.1. Implications

According to this study, pricing competition between drugstores and hospitals may cause drug shortage problems in China's pharmaceutical supply chains. The key reason for the problem is that the only goal for pharmaceutical retailers is to maximize the profits. Thus, the variation of factors such as ex-factory price, price sensitivity of demand and governmental discounts directly influence the satisfaction rates.

Fortunately, due to the centralized political system, there are three possible ways for China's government to increase the satisfaction rate of drugs. The first way is to set ex-factory price restrictions for certain drugs. Such policies are commonly adopted by municipal or provincial healthcare and security administrations [63]. The second way is to strengthen the medical insurance system and try to include more kinds of medications in it. Such policies could reduce the price sensitivity of demands which make the pharmaceutical suppliers more willing to produce [64]. The third way is to provide more discounts to hospitals. A typical example of it is the ongoing centralized drug bidding and purchasing system, in which China's government tries to integrate the purchasing activity of public hospitals to obtain a lower marginal cost, which could also relieve the pressure of drug shortage [65].

It is not strange that the disparity of drug price exists between drugstores and hospitals in China. Apart from drugs, hospitals also provide medical diagnosis and treatment, which often lead to a higher operation cost and thus a higher price than drugstores. But the abnormality of drugstores selling drugs four or five times more expensive than hospitals is worthy of note [15]. This situation is a reminder that some parts of China's pharmaceutical supply chains still lack regulations. Policies such as adjusting medical insurance systems, setting drug price restrictions or providing subsidies are necessary for people who are unwilling or unable to go to hospitals.

### 6.2. Suggestions for Future Research

There are many opportunities for future research. One extension is to include more hospitals and drugstores in the game. The existence and uniqueness of the equilibrium point may still be preserved, but the closed-form solution of optimal selling prices may no longer be found. The properties of the equilibrium may be analyzed with relevant optimal conditions. As hospitals can be regarded as social welfare institutions, a model in which hospitals are not allowed to be out of stock could also be examined.

Another interesting factor to consider is that the hospital and drugstore could sell multiple drugs that are used to treat a certain kind of disease. This game would then change to a multi-product model with substitution, for which [66] could provide some insights.

The problem could also be studied as a game with strategic customer behavior, meaning the advantage of the hospital can no longer be depicted by the differences between $A_h$ and $A_d$. The patients have three ways of buying drugs: go to a hospital for a doctor's advice and buy the drugs there; go directly to a drugstore without asking for any advice; or go to a hospital for advice and then buy the drugs at a drugstore. This model would be quite different from ours.

**Author Contributions:** Conceptualization, S.W., M.L.; Data curation, S.W., J.Z.; formal analysis, S.W., M.L., L.Z.; methodology, S.W., M.L., L.Z.; writing—original draft, S.W., M.L.; funding acquisition S.W., M.L.; investigation, J.Z., D.Z.; visualization, J.Z., D.Z.; writing—review & editing, D.Z., L.Z. All authors have read and agreed to the published version of the manuscript.

**Funding:** This research was funded by the Beidou 2.0 Scientific Research Program of Nanjing Polytechnic Institute (NO. NJPI-2022-YB-09) and Doctoral Research Program of Shenzhen Institute of Information Technology (NO. SZIIT2021KJ033).

**Institutional Review Board Statement:** Not applicable.

**Informed Consent Statement:** Not applicable.

**Data Availability Statement:** No new data were created or analyzed in this study.

**Acknowledgments:** The authors would like to thank the editors and three anonymous reviewers for their valuable comments and suggestions on an earlier version of this paper.

**Conflicts of Interest:** The authors declare no conflict of interest.

## Appendix A. Proof of Lemmas and Theorems

This section contains the proofs of all the lemmas and theorems in our work.

### Appendix A.1. Proof of Lemma 1

**Proof of Lemma 1.** Due to the symmetry of the drugstore and hospital, it is sufficient to show that the two $D_d$s satisfy Assumptions 3 and 4.

1. Linear Form:

   For Assumption 3, $\frac{\partial D_d}{\partial p_d} = -a_d < 0$, $\frac{\partial D_d}{\partial p_h} = b_d > 0$, $\frac{\partial D_d}{\partial A_d} = 1 > 0$.

   For Assumption 4, $e_d = \frac{a_d p_d}{A_d - a_d p_d + b_d p_h}$,

   thus $\frac{\partial e_d}{\partial p_d} = \frac{a_d(b_d p_h + A_d)}{(A_d - a_d p_d + b_d p_h)^2} > 0$ and $\frac{\partial e_d}{\partial p_h} = \frac{-a_d b_h p_d}{(A_d - a_d p_d + b_d p_h)^2} \leqslant 0$.

2. Logarithmic Form:

   For Assumption 3, we have $\frac{\partial D_d}{\partial p_d} = -\frac{a_d A_d A_h e^{-a_d p_d - a_h p_h}}{(A_d e^{-a_d p_d} + A_h e^{-a_h p_h})^2} < 0$,

   $\frac{\partial D_d}{\partial p_h} = \frac{a_h A_d A_h e^{-a_d p_d - a_h p_h}}{(A_d e^{-a_d p_d} + A_h e^{-a_h p_h})^2} > 0$ and $\frac{\partial D_d}{\partial A_d} = \frac{A_h e^{-a_d p_d - a_h p_h}}{(A_d e^{-a_d p_d} + A_h e^{-a_h p_h})^2} > 0$.

   For Assumption 4, we have $e_d = \frac{a_d A_h p_d e^{-a_h p_h}}{A_d e^{-a_d p_d} + A_h e^{-a_h p_h}}$,

   then $\frac{\partial e_d}{\partial p_d} = \frac{a_d A_h e^{-a_h p_h}(a_d A_d p_d e^{-a_d p_d} + A_d e^{-a_d p_d} + A_h e^{-a_h p_h})}{(A_d e^{-a_d p_d} + A_h e^{-a_h p_h})^2} > 0$,

   $\frac{\partial e_d}{\partial p_h} = -\frac{a_d a_h A_d A_h p_d e^{-a_d p_d - a_h p_h}}{(A_d e^{-a_d p_d} + A_h e^{-a_h p_h})^2} \leqslant 0$

   So the commonly used demand functions given in (1) and (2) satisfy assumptions 3 and 4. □

### Appendix A.2. Proof of Lemma 2

**Proof of Lemma 2.** Taking partial derivative on $\Pi_d$ with respect to $p_d$, we get $\frac{\partial \Pi_d(p_d, p_h, Q_d)}{\partial p_d} =$

$E_{\xi_d}[min\{Q_d, D_d \cdot \xi_d\}] + p_d \cdot \frac{\partial D_d}{\partial p_d} \cdot \int_0^{\frac{Q_d}{D_d}} t f_{\xi_d}(t) dt.$

Further taking partial derivative again, we have: $\frac{\partial^2 \Pi_d(p_d, p_h, Q_d)}{\partial(p_d)^2} = \frac{D_d}{p_d}\left(\frac{p_d^2}{D_d}\frac{\partial^2 D_d}{\partial p_d^2} - 2e_d\right)\int_0^{\frac{Q_d}{D_d}} t f_{\xi_d}(t)dt - \frac{D_d}{p_d}e_d^2\frac{Q_d^2}{D_d^2}f_{\xi_d}\left(\frac{Q_d}{D_d}\right)$.

As both of $-2e_d\frac{D_d}{p_d}$ and $-\frac{D_d}{p_d}e_d^2\frac{Q_d^2}{D_d^2}f_{\xi_d}\left(\frac{Q_d}{D_d}\right)$ are negative, it is sufficient to show $\frac{\partial^2 D_d}{\partial p_d^2} \leqslant 0$. According to Assumption 1, $\frac{\partial e_d}{\partial p_d} = -\frac{\partial^2 D_d}{\partial p_d^2}\frac{p_d}{D_d} + \frac{1}{p_d}e_d + \frac{1}{p_d}e_d^2 > 0$.

It means that $-\frac{\partial^2 D_d}{\partial p_d^2}\frac{p_d^2}{D_d} + e_d + e_d^2 > 0$ will always hold for any $e_d$, which implies $\frac{\partial^2 D_d}{\partial p_d^2} \leqslant 0$.

So we have $\frac{\partial^2 \Pi_d(p_d, p_h, Q_d)}{\partial(p_d)^2} < 0$ which indicates that $\Pi_d$ is a concave function with respect to $p_d$. Note that we have not use any special properties which differ the drugstore from the hospital, the conclusions which we have drawn on the drugstore will also be true on the hospital. So $\Pi_h$ is also a concave function with respect to $p_h$. □

*Appendix A.3. Proof of Lemma 3*

**Proof of Lemma 3.** We transform the drugstore's payoff function $\Pi_d(p_d, p_h, Q_d^*) = 4D_d p_d \sigma_d$ $(1 - \frac{\sigma_d c}{p_d})(1 - \frac{c}{p_d})$ to the logarithmic form $\ln[\Pi_d(p_d, p_h, Q_d^*)]$.

As $\ln x$ is a continuous monotonic increasing function, this transformation will not change the optimal solution of the game. It is equivalent to show that the new game with the payoff function $\ln[\Pi_d(p_d, p_h, Q_d^*)]$ is supermodular.

It is obvious that $\ln[\Pi_d(p_d, p_h, Q_d^*)]$ is twice continuously differentiable, we will prove that $\frac{\partial^2 \ln[\Pi_d(p_d, p_h, Q_d^*)]}{\partial p_d \partial p_h}$ is non-negative. Note

$$\frac{\partial \ln[\Pi_d(p_d, p_h, Q_d^*)]}{\partial p_d} = -\frac{e_d}{p_d} + \frac{1}{p_d - c} + \frac{1}{p_d - \sigma_d c} - \frac{1}{p_d}. \tag{A1}$$

Thus

$$\frac{\partial^2 \ln[\Pi_d(p_d, p_h, Q_d^*)]}{\partial p_d \partial p_h} = -\frac{1}{p_d}\cdot\frac{\partial e_d}{\partial p_h}. \tag{A2}$$

According to Assumption 4.2, we know that $\frac{\partial^2 \ln[\Pi_d(p_d, p_h, Q_d^*)]}{\partial p_d \partial p_h}$ and $\frac{\partial^2 \ln[\Pi_d(p_d, p_h, Q_d^*)]}{\partial p_d \partial p_h}$ are non-negative. Repeat this procedure again we get

$$\frac{\partial^2 \ln[\Pi_h(p_d, p_h, Q_h^*)]}{\partial p_h \partial p_d} = -\frac{1}{p_h}\cdot\frac{\partial e_h}{\partial p_d} \geqslant 0. \tag{A3}$$

Hence both of the two player's payoffs are supermodular, which by definition, proves supermodularity of the game. □

*Appendix A.4. Proof of Theorem 1*

**Proof of Theorem 1.** Cachon and Netessine have stated that in a supermodular game, there exists at least one Nash Equilibrium (Theorem 3, [61]). According to Assumption 2, each of the two players' decision varies in a closed interval, hence the strategy space of the game is a convex compact set in $\mathbb{R}^2$. By Lemma 3, we know this game is supermodular, so there exists at least one Nash Equilibrium in it. □

*Appendix A.5. Proof of Lemma 4*

**Proof of Lemma 4.** By Assumption 4.1 we know $\frac{\partial e_d}{\partial p_d} > 0$, thus $(-e_d + 1)$ is monotonic decreasing with $p_d$.

It is obvious that $\frac{c}{p_d}$ is also monotonic decreasing with $p_d$ and $\frac{c}{p_d} > 0$. Next we will show that $\frac{-2\sigma_d c/p_d + 1 + \sigma_d}{(-2\sigma_d c/p_d + 1 + \sigma_d)^2 - (1-\sigma_d)^2}$ is also monotonic decreasing with $p_d$ and non-negative.

Let $h(p_d) = \frac{-2\sigma_d c/p_d + 1 + \sigma_d}{(-2\sigma_d c/p_d + 1 + \sigma_d)^2 - (1-\sigma_d)^2}$. If $p_d = c$, we would get $\Pi_d(p_d, p_h, Q_d^*) = D_d p_d \int_0^{F_{\xi_d}^{-1}(0)} t f_{\xi_d}(t) dt = 0$, which is meaningless, thus we could only consider the situation that $p_d > c$.

As $\sigma_d \in [0,1]$, we have $-2\sigma_h \phi c/p_h + 1 + \sigma_h > 1 - \sigma_d \geqslant 0$.

Thus $h(p_d)$ can be rewritten as $h(p_d) = \frac{1}{(-2\sigma_d c/p_d^* + 1 + \sigma_d) - \frac{(1-\sigma_d)^2}{-2\sigma_d c/p_d^* + 1 + \sigma_d}}$.

Note that $-2\sigma_d c/p_d^* + 1 + \sigma_d$ is monotonic increasing with $p_d$, which directly derives that $\frac{-(1-\sigma_d)^2}{-2\sigma_d c/p_d^* + 1 + \sigma_d}$ is also monotonic increasing with $p_d$, we can conclude that $h(p_d)$ is monotonic decreasing with $p_d$. Furthermore, we have $\lim\limits_{p_d \to \infty} h(p_d) = \frac{1+\sigma_d}{4\sigma_d} > 0$.

Thus $h(p_d)$ is monotonic decreasing with $p_d$ and non-negative.

Recall that $g_d(p_d) = -e_d + 1 + \frac{2c}{p_d} \cdot \frac{-2\sigma_d c/p_d + 1 + \sigma_d}{(-2\sigma_d c/p_d + 1 + \sigma_d)^2 - (1-\sigma_d)^2}$, we could rewrite it as $g_d(p_d) = -e_d + 1 + \frac{2c}{p_d} \cdot h(p_d)$.

Taking derivative on $g_d(p_d)$ with $p_d$, we get $\frac{\partial g_d(p_d)}{\partial p_d} = -\frac{\partial e_d}{\partial p_d} - \frac{2c}{p_d^2} \cdot h(p_d) + \frac{2c}{p_d} \cdot \frac{\partial h(p_d)}{\partial p_d} < 0$, which means $g_d(p_d)$ is strictly monotonic decreasing with $p_d$.

Further taking derivative on $g_d(p_d)$ with $p_h$, we get $\frac{\partial g_d(p_d)}{\partial p_h} = -\frac{\partial e_d}{\partial p_h}$, which is by Assumption 4.2, non-negative. Thus $g_d(p_d)$ is non-decreasing with $p_h$.

Similarly, we could see $\frac{\partial g_h(p_h)}{\partial p_h} = -\frac{\partial e_h}{\partial p_h} - \frac{2\phi c}{p_h^2} \cdot h(p_h) + \frac{2\phi c}{p_h} \cdot \frac{\partial h(p_h)}{\partial p_h} < 0$ and $g_d(p_d) = -\frac{\partial e_h}{\partial p_d} \geqslant 0$.

Thus $g_h(p_h)$ is non-decreasing with $p_h$, and strictly monotonic decreasing with $p_d$.

For the drugstore, by Theorem 1 and definition of Nash equilibrium we have known that the best response function exists and must satisfy $g_d(p_d) = 0$. As we have proved above, $g_d(p_d)$ could only cross 0 one time, thus the best response function of the drugstore $r_d^*(p_h)$ can be uniquely determined by solving $g_d(p_d) = 0$.

The same statement can be made on that of the hospital. □

*Appendix A.6. Proof of Lemma 5*

**Proof of Lemma 5.** We prove the necessity and sufficiency, separately.

- Necessity
  According to definition, a function $f(x)$ is called quasi-concave if $f(tx_1 + (1-t)x_2) \geqslant \min(f(x_1), f(x_2))$ holds for all $t \in (0,1)$.

  Assume $f$ is neither monotonic nor first non-decreasing and then non-increasing. It means that $f(x)$ is either first decreasing and increasing or first non-decreasing and then non-increasing but at last decreasing and increasing again. As $f(x)$ is twice continuously differentiable, it is equivalent to say $f'(x)$ will cross 0 more than once. Let $a_1$, $a_2$ be the two adjacent crossing points that satisfy $f'(a_i) = 0$. There exist a $\delta > 0$, for any $0 < \varepsilon < \delta$, we have $f'(a_i + \varepsilon) \cdot f'(a_i - \varepsilon) < 0$, where $i = 1, 2$. Without loss of generality, we assume $a_1 < a_2$.

  1. If $f(a_1) < f(a_2)$
     Because $f(a_1) = f(a_2) = 0$, we know that there exist a $\delta_1 > 0$, such that for all $x \in [a_1 - \delta_1, a_1]$, $f(x)$ is strictly monotonic decreasing and all $x \in [a_1, a_1 + \delta_1]$, $f(x)$ is strictly monotonic increasing. This means that $f(x)$ is convex on $[a_1 - \delta_1, a_1 + \delta_1]$, which contracts the quasi-concavity of $f(x)$.
  2. If $f(a_1) > f(a_2)$
     Similarly, we could say that there exist a $\delta_2 > 0$, such that $f(x)$ is convex on $[a_2 - \delta_2, a_2 + \delta_2]$, which also contracts the quasi-concavity of $f(x)$.

  Thus our assumption is wrong, which means $f(x)$ is either monotonic or first non-decreasing and then non-increasing.

- Sufficiency
  Take any $x_1$, $x_2$, without loss of generality, we assume $x_1 < x_2$.
  1. If $f(x)$ is monotonic increasing, we have $f(x_1) \leqslant f(x_2)$, then $f(tx_1 + (1-t)x_2) \geqslant f(x_1) = \min(f(x_1), f(x_2))$, thus $f(x)$ is quasi-concave.
  2. If $f(x)$ is monotonic decreasing, we have $f(x_1) \geqslant f(x_2)$, then $f(tx_1 + (1-t)x_2) \geqslant f(x_2) = \min(f(x_1), f(x_2))$, thus $f(x)$ is quasi-concave.
  3. If $f(x)$ is first non-decreasing and then non-increasing. Let $x_0$ denote the turning point of $f(x)$, that is, for $x < x_0$, $f(x)$ is non-decreasing and $x > x_0$, $f(x)$ is non-increasing.
     (a) If $x_1 < x_2 \leqslant x_0$, we have $f(x_1) \leqslant f(x_2)$, then $f(tx_1 + (1-t)x_2) \geqslant f(x_1) = \min(f(x_1), f(x_2))$.
     (b) If $x_0 < x_1 < x_2$, we have $f(x_1) \geqslant f(x_2)$, then $f(tx_1 + (1-t)x_2) \geqslant f(x_2) = \min(f(x_1), f(x_2))$.
     (c) If $x_1 < x_0 < x_2$, we will discuss the position of $x_3 = tx_1 + (1-t)x_2$.
        i. If $x_1 < x_3 \leqslant x_0 < x_2$, we have $f(x_3) \geqslant f(x_1)$, which means $f(tx_1 + (1-t)x_2) \geqslant f(x_1) = \min(f(x_1), f(x_2))$.
        ii. If $x_1 < x_0 < x_3 < x_2$, we have $f(x_3) \geqslant f(x_2)$, which means $f(tx_1 + (1-t)x_2) \geqslant f(x_2) = \min(f(x_1), f(x_2))$.

  As discussed above, the sufficiency is proved.

  Thus a twice continuously differentiable function $f$ defined on $\mathbb{R}$, is quasi-concave if and only if it is monotonic or first non-decreasing and then non-increasing. $\square$

*Appendix A.7. Proof of Lemma 6*

**Proof of Lemma 6.** We prove that $\Pi_d(p_d, p_h, Q_d^*)$ is quasi-concave in $p_d$. The proof of $\Pi_h$ in $p_h$ is almost the same.

$\Pi_d(p_d, p_h, Q_d^*)$ is of course twice continuously differentiable, and we have shown in Section 4.2.2 that $\frac{\partial \Pi_d(p_d, p_h, Q_d^*)}{\partial p_d} = 0$ is equivalent to $g_d(p_d) = 0$. By Lemma 4 we know that for any given $p_h$, it has an unique solution $p_d = r_d^*(p_h)$. Thus the deviation of $\Pi_d(p_d, p_h, Q_d^*)$ can cross 0 only once.

Further notice that $\lim\limits_{p_d \to c_+} \frac{\partial \Pi_d(p_d, p_h, Q_d^*)}{\partial p_d} = 4D_d \sigma_d(1 - \sigma_d) \geqslant 0$.

Thus $\Pi_d(p_d, p_h, Q_d^*)$ is first non-decreasing and then non-increasing in $p_d$. By Lemma 5 $\Pi_d(p_d, p_h, Q_d^*)$ is quasi-concave in $p_d$.

Similarly, $\Pi_h(p_d, p_h, Q_h^*)$ is quasi-concave in $p_h$. $\square$

*Appendix A.8. Proof of Theorem 2*

**Proof of Theorem 2.** Provided that the strategy space of the game is convex and the payoff functions are quasi-concave. Netessine and Cathon's index theory approach theorem (Theorem 7, [62]) tells us that if the determinant of Hessian $H = \begin{vmatrix} \frac{\partial^2 \Pi_d}{\partial p_d^2} & \frac{\partial^2 \Pi_d}{\partial p_d \partial p_h} \\ \frac{\partial^2 \Pi_h}{\partial p_h \partial p_d} & \frac{\partial^2 \Pi_h}{\partial p_h^2} \end{vmatrix}$ is positive whenever $\frac{\partial \Pi_d}{\partial p_d} = 0$ and $\frac{\partial \Pi_h}{\partial p_h} = 0$ are satisfied simultaneously, the game will have a unique Nash equilibrium.

As $\ln f(x)$ and $f(x)$ have the same maximum point, the logarithmic transformation is order-preserving. We will prove it on the logarithmic form $\ln(\Pi_d)$ and $\ln(\Pi_h)$.

Substituting $\frac{\partial \ln \Pi_d}{\partial p_d} = 0$ and $\frac{\partial \ln \Pi_h}{\partial p_h} = 0$ into the determinant of Hessian and after some simplifications, we have

$$det H = N_d \frac{\partial e_h}{\partial p_h} \frac{1}{p_h} + N_h \frac{\partial e_d}{\partial p_d} \frac{1}{p_d} + N_d N_h + \frac{1}{p_d p_h} \left( \frac{\partial e_d}{\partial p_d} \frac{\partial e_h}{\partial p_h} - \frac{\partial e_h}{\partial p_d} \frac{\partial e_d}{\partial p_h} \right), \tag{A4}$$

where $N_p = \frac{c}{p_d(p_d - c)^2} + \frac{\sigma_d c}{p_d(p_d - \sigma_d c)^2}$ and $N_h = \frac{\phi c}{p_h(p_h - \phi c)^2} + \frac{\sigma_h \phi c}{p_h(p_h - \sigma_h \phi c)^2}$.

It is clear that $N_p$ and $N_h$ are positive, so by Assumption 1 the first three parts of Equation (A4) are positive. By Assumption 5 we have $\frac{\partial e_d}{\partial p_d} \geqslant -\frac{\partial e_d}{\partial p_h} \geqslant 0$ and $\frac{\partial e_h}{\partial p_h} \geqslant -\frac{\partial e_h}{\partial p_d} \geqslant 0$.

So $\frac{\partial e_d}{\partial p_d} \frac{\partial e_h}{\partial p_h} \geqslant \frac{\partial e_h}{\partial p_d} \frac{\partial e_d}{\partial p_h}$, which means the forth part of Equation (A4) is non-negative.

Thus, we finally obtain $det H > 0$, as a result, the game has a unique Nash equilibrium. Furthermore, the Nash equilibrium point $(p_d^*, p_h^*)$ must satisfy condition (9), so it can be uniquely solved by condition (9). □

*Appendix A.9. Proof of Theorem 3*

**Proof of Theorem 3.** We have calculated in Section 4.3 that $Q_d^* = (A_d - a_d p_d^* + b_d p_h^*)(-\frac{2c}{p_d^*} + 2)$, together with Equation (10) we have $s_d^* = 1 - \frac{c}{p_d^*} = 1 - \frac{4 a_d a_h - b_d b_h}{2 a_d a_h + a_h b_d \phi + \frac{2 a_h A_d + b_d A_h}{c}}$, which is monotonically decreasing with $c$ and increasing with $\phi$.

Similarly, we have $s_h^* = 1 - \frac{\phi c}{p_h^*} = 1 - \frac{4 a_d a_h - b_d b_h}{2 a_d a_h + \frac{a_d b_h}{\phi} + \frac{2 a_d A_h + b_h A_d}{\phi c}}$, which is monotonically decreasing with $c$ and $\phi$. □

*Appendix A.10. Proof of Theorem 4*

**Proof of Theorem 4.** From Equation (11) we have $p_d^* = \frac{2c + \frac{k \phi c + 2 A_d}{l} + \frac{k A_h}{l^2}}{4 - \frac{k^2}{l^2}}$.

It is obvious that $4 - \frac{k^2}{l^2}$ is increasing with $l$ and $2c + \frac{k \phi c + 2 A_d}{l} + \frac{k A_h}{l^2}$ is decreasing with $l$, so $p_d^*$ is monotonically decreasing with $l$. We also have $2l^2 c + lk\phi c + 2l A_d + k A_h$ is increasing with $k$ and $4l^2 - k^2$ is decreasing with $k$, so $p_d^*$ is monotonically increasing with $k$.

Similarly, we have $p_h^*$ is monotonically decreasing with $l$ and increasing with $k$.

Note that $s_d^* = 1 - \frac{c}{p_d^*}$, which is monotonically increasing with $p_d^*$, so $s_d^*$ is also monotonically decreasing with $l$ and increasing with $k$.

Similarly, we have $s_h^*$ is monotonically decreasing with $l$ and increasing with $k$. □

*Appendix A.11. Proof of Theorem 5*

**Theorem 5.** From Equation (12) we have $p_d^* = \frac{2c}{3} + \frac{h \phi c + 2 A_d + A_h}{3d}$, which is monotonically decreasing with $d$ and increasing with $h$. Similarly, we have $p_h^* = \frac{2 \phi c}{3} + \frac{dc + A_d + 2 A_h}{3h}$, which is monotonically decreasing with $h$ and increasing with $d$.

Note that $s_d^* = 1 - \frac{c}{p_d^*}$, which is monotonically increasing with $p_d^*$, so $s_d^*$ is monotonically decreasing with $d$ and increasing with $h$.

Similarly, we have $s_h^*$ is monotonically decreasing with $h$ and increasing with $d$. □

**Appendix B. Equation Derivations**

This appendix contains the derivations of rerlated equations.

*Appendix B.1. The Derivation of Equations (7) and (8)*

As the derivation of Equations (7) and (8) are almost the same, we will only derive the drugstore's part.

Because $\frac{\partial \int_0^{F_{\xi_d}^{-1}(\rho_d)} t f_{\xi_d}(t) dt}{\partial p_d} = F_{\xi_d}^{-1}(\rho_d) \cdot f_{\xi_d}(F_{\xi_d}^{-1}(\rho_d)) \cdot \frac{\partial F_{\xi_d}^{-1}(\rho_d)}{\partial p_d} = \frac{c_d}{(p_d)^2} \cdot F_{\xi_d}^{-1}(\rho_d)$.

We have

$$\frac{\partial \Pi_d(p_d, p_h, Q_d^*)}{\partial p_d}$$

$$= [\frac{\partial D_d(p_d, p_h, A_d)}{\partial p_d} p_d + D_d] \int_0^{F_{\xi_d}^{-1}(\rho_d)} t f_{\xi_d}(t) dt + D_d p_d \cdot \frac{\partial \int_0^{F_{\xi_d}^{-1}(\rho_d)} t f_{\xi_d}(t) dt}{\partial p_d}$$

$$= D_d \left[ (-e_d + 1) \cdot \int_0^{F_{\xi_d}^{-1}(\rho_d)} t f_{\xi_d}(t) dt + \frac{c_d}{p_d} \cdot F_{\xi_d}^{-1}(\rho_d) \right].$$

Repeating this procedure again, we will see

$$\frac{\partial \Pi_h(p_d, p_h, Q_h^*)}{\partial p_h} = D_h \left[ (-e_h + 1) \cdot \int_0^{F_{\xi_h}^{-1}(\rho_h)} t f_{\xi_h}(t) dt + \frac{\phi c}{p_h} \cdot F_{\xi_h}^{-1}(\rho_h) \right].$$

*Appendix B.2. The Derivation of Equation (A4)*

In the proof of Lemma 3, we have worked out (see Equation (A2) and (A3)):
$\frac{\partial^2 \ln[\Pi_d(p_d,p_h,Q_d^*)]}{\partial p_d \partial p_h} = -\frac{1}{p_d} \cdot \frac{\partial e_d}{\partial p_h}$ and $\frac{\partial^2 \ln[\Pi_h(p_d,p_h,Q_h^*)]}{\partial p_h \partial p_d} = -\frac{1}{p_h} \cdot \frac{\partial e_h}{\partial p_d}$.
We also obtain (see equation (A1)):
$\frac{\partial \ln[\Pi_d(p_d,p_h,Q_d^*)]}{\partial p_d} = -\frac{e_d}{p_d} + \frac{1}{p_d-c} + \frac{1}{p_d-\sigma_d c} - \frac{1}{p_d}$,
$\frac{\partial^2 \ln[\Pi_d(p_d,p_h,Q_d^*)]}{\partial p_d^2} = \frac{1+e_d}{p_d^2} - \frac{1}{(p_d-c)^2} - \frac{1}{(p_d-\sigma_d c)^2} - \frac{\partial e_d}{\partial p_d} \cdot \frac{1}{p_d}$.
Similarly, we have $\frac{\partial^2 \ln[\Pi_h(p_d,p_h,Q_h^*)]}{\partial p_h^2} = \frac{1+e_h}{p_h^2} - \frac{1}{(p_h-\phi c)^2} - \frac{1}{(p_h-\sigma_h \phi c)^2} - \frac{\partial e_h}{\partial p_h} \cdot \frac{1}{p_h}$.
Let $M_p = \frac{1}{(p_d-c)^2} + \frac{1}{(p_d-\sigma_d c)^2}$ and $M_h = \frac{1}{(p_h-\phi c)^2} + \frac{1}{(p_h-\sigma_h \phi c)^2}$.
The determinant of the Hessian in logarithmic form is

$$\frac{\partial^2 \ln\Pi_d}{\partial p_d^2} \cdot \frac{\partial^2 \ln\Pi_d}{\partial p_h^2} - \frac{\partial^2 \ln\Pi_d}{\partial p_d \partial p_h} \cdot \frac{\partial^2 \ln\Pi_d}{\partial p_h \partial p_d} = \left[ \frac{M_d}{p_h} - \frac{1+e_d}{p_h p_d^2} \right] \cdot \frac{\partial e_h}{\partial p_h} + \left[ \frac{M_h}{p_d} - \frac{1+e_h}{p_d p_h^2} \right] \cdot \frac{\partial e_d}{\partial p_d}$$

$$+ \left[ M_d - \frac{1+e_d}{p_d^2} \right] \left[ M_h - \frac{1+e_h}{p_h^2} \right] + \frac{1}{p_d p_h} \left( \frac{\partial e_d}{\partial p_d} \frac{\partial e_h}{\partial p_h} - \frac{\partial e_h}{\partial p_d} \frac{\partial e_d}{\partial p_h} \right).$$

When $\frac{\partial \ln\Pi_d}{\partial p_d} = 0$ and $\frac{\partial \ln\Pi_h}{\partial p_h} = 0$ are satisfied simultaneously, we have $\frac{1+e_d}{p_d^2} = \frac{1}{p_d(p_d-c)} + \frac{1}{p_d(p_d-\sigma_d c)}$ and $\frac{1+e_h}{p_h^2} = \frac{1}{p_h(p_h-\phi c)} + \frac{1}{p_h(p_h-\sigma_h \phi c)}$.

Thus when the first derivative of the two payoff functions are equal to zero, we have $M_d - \frac{1+e_d}{p_d^2} = \frac{c}{p_d(p_d-c)^2} + \frac{\sigma_d c}{p_d(p_d-\sigma_d c)^2}$ and $M_h - \frac{1+e_h}{p_h^2} = \frac{\phi c}{p_h(p_h-\phi c)^2} + \frac{\sigma_h \phi c}{p_h(p_h-\sigma_h \phi c)^2}$.

Using $N_d$ and $N_h$ to denote $\frac{c}{p_d(p_d-c)^2} + \frac{\sigma_d c}{p_d(p_d-\sigma_d c)^2}$ and $\frac{\phi c}{p_h(p_h-\phi c)^2} + \frac{\sigma_h \phi c}{p_h(p_h-\sigma_h \phi c)^2}$, respectively, we finally obtain

$$det H = N_d \frac{\partial e_h}{\partial p_h} \frac{1}{p_h} + N_h \frac{\partial e_d}{\partial p_d} \frac{1}{p_d} + N_d N_h + \frac{1}{p_d p_h} \left( \frac{\partial e_d}{\partial p_d} \frac{\partial e_h}{\partial p_h} - \frac{\partial e_h}{\partial p_d} \frac{\partial e_d}{\partial p_h} \right).$$

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
