# Peer review of "Pharmaceutical Supply Chain in China: Pricing and Production Decisions with Price-Sensitive and Uncertain Demand"

_sustainability, doi:10.3390/su14137551_

Round 1

Reviewer 1 Report

This is an interesting paper, however I do not see any relevance with the Sustainability section: Economic and Business Aspects of Sustainability. In my opinion this manuscript is out of journal's scope. Maybe authors will consider to send this manuscript to Mathematics?

Also, it is not formatted according to Sustainability editorial requirements.

The methodology section with all formulas is very difficult to understand. It should be presented in more clear way.

Reviewer 2 Report

The article is an interesting one. The article covers a wide set of disciplines. The following comments will help improve the manuscript.

1. The introduction section is reasonable, written well, but very narrowly focussed, therefore the authors must check to make it more relevant to the journal.

2.  The research question(s) are not clear, if the authors can add research objectives following research contributions, then it will increase readability.

3. Research assumptions need more theoretical base. Author(s) should link the assumptions to existing literature.

4. It would be nice to see some kind of conceptual framework for this research.

5. I would suggest refining your research methodology. At the moment, rather limited discussion on research philosophy.

6. Critically analyse different theoretical approaches to the research problem

7. Justify the solution method selected in terms of the research objectives

8. Strengthen the motivation and discussion parts substantially.

9. Demonstrate adequately that your solution findings have been logically derived and that conclusions, solutions/recommendations are fully supported by evidence

10. While Pharma SC is still a budding topic, still one can find many research papers in other reputed journals as well. I'd suggest referring to the below academic papers in your manuscript:

·        Sharma et al. 2022. The Impact of Environment Dynamism on Low-Carbon Practices and Digital Supply Chain Networks to Enhance Sustainable Performance: An Empirical Analysis. Business Strategy and the Environment.

·        Jraisat et al. 2021. Triads in Sustainable Supply-Chain Perspective: Why is a Collaboration Mechanism Needed?, International Journal of Production Research.

·        Jaeger et al. 2021. Identification of environmental supply chain bottlenecks: A case study of the Ethiopian Healthcare Supply Chain. Management of Environmental Quality: An International Journal.

·        Upadhyay, A. 2020. Antecedents of green supply chain practices in developing economies. Management of Environmental Quality: An International Journal.

11.     The authors must check all references carefully, there are some references are cited in the text and vice versa.

Reviewer 3 Report

I have read the manuscript thoroughly and found that the study is about the supply chain of pharmaceutical companies in China, where the authors have applied game theory to study the price competition among various variables available in the market. The article is interesting with respect to its area of research. There are few improvements which can be considered by the authors towards its possible publication of the article. These improvements are :

I was looking for the motivation and the research gap for which you have brought this study but its missing. I think you should highlight that.

Highlight a clear presentation of your research methodology. 

How your research is different from other studies in this area? What are the uniqueness? Cite 

In the conclusion section you mentioned as 

Most previous studies of pharmaceutical supply chains have focused on network design, capacity planning, product portfolio selection, scheduling, and inventory management, and they have failed to consider price competition between retailers.  What are these studies cite it and what they talk about ?

Write implications of your study.

References section in the end of the paper are poorly presented. Please follow APA or specified style by the journal for the referencing. 

All the best.

Round 2

Reviewer 1 Report

I have checked this paper once again and it seems that authors applied all suggestes changes. I have no further remarks.

Reviewer 2 Report

Thank you for your hard work. The paper looks better now.